# Prosocial correlates of transformative experiences at secular multi-day mass gatherings

Daniel A. Yudkin[1✉], Annayah M. B. Prosser[2], S. Megan Heller[3], Kateri McRae[4], Aleksandr Chakroff[6] & M. J. Crockett [5✉]

Humans have long sought experiences that transcend or change their sense of self. By weakening boundaries between the self and others, such transformative experiences may lead to enduring changes in moral orientation. Here we investigated the psychological nature and prosocial correlates of transformative experiences by studying participants before ($n = 600$), during ($n = 1217$), 0–4 weeks after ($n = 1866$), and 6 months after ($n = 710$) they attended a variety of secular, multi-day mass gatherings in the US and UK. Observations at 6 field studies and 22 online followup studies spanning 5 years showed that self-reported transformative experiences at mass gatherings were common, increased over time, and were characterized by feelings of universal connectedness and new perceptions of others. Participants' circle of moral regard expanded with every passing day onsite—an effect partially mediated by transformative experience and feelings of universal connectedness. Generosity was remarkably high across sites but did not change over time. Immediately and 6 months following event attendance, self-reported transformative experience persisted and predicted both generosity (directly) and moral expansion (indirectly). These findings highlight the prosocial qualities of transformative experiences at secular mass gatherings and suggest such experiences may be associated with lasting changes in moral orientation.

[1] Department of Psychology, University of Pennsylvania, 3720 Walnut St., Philadelphia, PA 19104, USA. [2] Department of Psychology, University of Bath, Claverton Down, Bath, UK. [3] Department of Anthropology, University of California, Los Angeles, CA 90095, USA. [4] Department of Psychology, University of Denver, 2155S Race St., Denver, CO 80210, USA. [5] Department of Psychology, Yale University, 2 Hillhouse Avenue, New Haven, CT 06511, USA. [6] Unaffiliated: Aleksandr Chakroff. ✉email: dyudkin@sas.upenn.edu; mj.crockett@yale.edu

The sociologist Emile Durkheim coined the term "collective effervescence" to describe the feelings of self-transcendence that often arise at mass gatherings such as festivals, pilgrimages, and collective rituals[1]. Testimonial accounts from attendees of mass gatherings suggest that such self-transcendent experiences may be epistemically and personally transformative, resulting in lasting changes to the self[2,3]. The philosopher L.A. Paul articulates two key components of transformative experiences: they provide new knowledge that is impossible to attain without having the experience, and they produce changes in personal values and priorities that cannot be anticipated[4]. Here, we investigate the psychological qualities of transformative experiences at secular mass gatherings and test the possibility that such experiences are associated with enduring changes in moral orientation.

Support for our hypothesis comes from research showing that collective gatherings such as rituals[5], ceremonies[6], raves[7], and sporting events[8,9] are associated with feelings of self-expansion and "group identity fusion"[10–13] a psychological state characterized by intense feelings of merging or oneness between the self and the group[14,15]. This phenomenon is well-documented across cultures[16] and is associated with endorsement of group values[17,18], enhanced group loyalty[6,19], increased generosity[20–26], and cooperation[27–30]. Such prosocial changes are thought to emerge from psychological processes whose original adaptive function lay in promoting the survival of the group at the expense of the self[31]. For example, anthropological accounts suggest that humans throughout history have often engaged in ritualistic behaviors (e.g., ceremonial dancing) that amplify identity fusion before performing acts of extreme self-sacrifice[32].

Past work on mass gatherings and identity fusion has tended to focus on how such experiences can amplify prosocial feelings and behavior directed toward the members of one's own social group. Here, by contrast, we investigated the potential for transformative experiences at mass gatherings to expand the boundaries of one's moral circle beyond the group to include all of humanity[33–35]. Recent research suggests that experiences of self-transcendent emotions are associated with increased identification with all humanity—i.e., increased concern for all other human beings—and motivations to help distant others[36,37]. Other work has shown that such experiences are associated with heightened feelings of social connectedness[38]. Building on this and other recent theoretical work[4], we conceptualized transformative experiences at mass gatherings as precipitating events that may lead to changes in values and behavior characterized by an increased sense of connection to other human beings and an expanded moral circle. Thus, we tested whether self-reported transformative experiences at secular mass gatherings were accompanied by increased feelings of universal connectedness, and investigated whether such experiences would be associated with enduring changes in generosity and moral expansion.

For practical and ethical reasons, it is difficult to generate in the lab the kinds of intensely transformative experiences that people report having at mass gatherings. Previous research on such experiences has relied on retrospective surveys that prompt participants to report on experiences they had in the past[7,25], yet such approaches are subject to the vagaries of personal recollection. We sought to build on this work by studying the psychological qualities of transformative experiences as they occurred. To do this, we adopted a lab-in-the-field approach in which we collected data from participants as they attended one of six secular multi-day mass gatherings across five field sites in the US and the UK (see Fig. 1 and Table 1). We define secular mass gatherings as events with a total attendance greater than five hundred[39] and no explicitly religious component. We focused on secular events in order to ensure that any effects observed were

not the result of explicit reference to the divine, which past research has shown to foster self-transcendent experiences[40] but is not the focus of this investigation. We also focused our research efforts on non-political events in order to avoid the potential confounding effects of specific ideological messages on prosocial attitudes and behavior.

Mass gatherings are highly immersive social experiences that may strip away belief systems and aspects of the self-concept like layers of an onion. Such experiences may thus not have their impact immediately upon arrival but instead take many days to impart their full effect[41]. For this reason, a central question in our research was how transformative experiences and their putative prosocial correlates unfold over time both during the events themselves and in the weeks and months after attendance. We thus focused our research on events with a total duration of three or more days and employed a data collection strategy that sampled participants at different timepoints within each event (see Fig. 1 and SOM 1.2). This allowed us to use time-at-event as a predictor in our models while controlling for key characteristics in our population sample. For example, while it is possible that people who attend mass gatherings are on average more likely to report transformative experiences or display prosocial attitudes and behavior, we were able to study the temporal dynamics of transformative experience and its prosocial correlates with our study population while holding these factors constant by having all participants complete our study at a relatively random timepoint during each event (see "Methods" for more information on our sampling strategy). Further, by sampling participants in the weeks and months following mass gathering attendance, we were additionally able to examine whether correlations between self-reported transformative experience and prosocial attitudes and behavior evolve over time.

We designed a variety of experimental procedures and methods appropriate for the mass gathering context, including measures of generosity and moral expansion that did not rely on the use of money, which was prohibited at some of our field sites (see SOM 4 for full descriptions of measures and verbatim participant instructions). We also collected detailed measures of participants' use of psychoactive substances, as past work has implicated psychedelic substance use in transformative experience and prosocial behavior[7,42,43]. In related work, we have examined the emotional consequences of psychedelic substance use at mass gatherings[44], finding that their use is associated with increased positive mood—an effect mediated by transformative experience and universal connectedness. Here, we sought to examine the nature and prosocial correlates of transformative experiences that do not necessarily arise from psychedelic substance use. To this end, we controlled for substance use in all analyses of onsite data, and compared the psychological qualities of transformative experience that arose in the presence versus absence of psychedelic substances (see SOM 1.2 for further details).

Finally, we investigated whether the "set and setting" of transformative experiences is associated with the nature of prosocial change[45,46]. While these concepts where originally conceived to understand the psychological consequences of psychedelic substance use[47], we surmised they may have similarly meaningful consequences for mass gathering participation alone. Related to "set" (i.e., mindset), it is known that desires and expectations can have powerful effects on subsequent experiences, for example in the placebo effect[48] and self-fulfilling prophecies[49]. Conversely, the absence of a desire or expectation for change may preclude actual change, a phenomenon known in clinical psychology as "client resistance"[50]. Accordingly, we sought to characterize the relationships between expecting and desiring transformative experiences, on the one hand, and actual self-reported transformative experiences on the other,

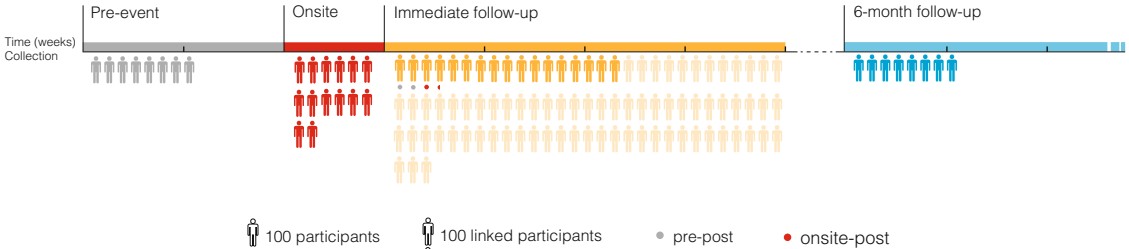

**Fig. 1 Stages of data collection.** Each icon represents 100 participants. Pre-event participants were recruited via newsletters sent out to all Burning Man attendees 0–2 weeks before attendance. Onsite participants were recruited by teams of experimenters hosting a booth with a sign that read "Play Games for Science." Immediate follow-up participants were recruited 0–4 weeks after attendance via e-mails sent to onsite participants (dark yellow). An additional supplementary sample was collected through the Burning Man Census (light yellow). Participants in the 6-month follow-up were recruited via e-mails sent to onsite participants and newsletters sent to all mass gathering participants 6 months following the event. Participants in the immediate follow-up dataset for whom we also have pre-event data ("pre-post") are marked with gray dots. Participants in the immediate follow-up dataset for whom we also have onsite data ("onsite-post") are marked with red dots. Source data are provided as a Source Data file.

**Table 1 Summary of field sites.**

|  | Burning Man | | | Burning Nest | Latitude | Dirty Bird Campout | Lightning in a bottle | Total |
|---|---|---|---|---|---|---|---|---|
| Total number attendees | ~70,000 | | | ~500 | ~40,000 | ~3000 | ~20,000 | |
| Location | US | | | UK | UK | US | US | |
| Gift economy | ✓ | | | ✓ | | | | |
| Event date | Aug '15 | Aug '16 | Aug '18 | June '16 | July '16 | Nov '17 | May '17 | |
| Cross-sectional | | | | | | | | |
| Pre-event (N) | 243 | 357 | | | | | | 600 |
| Onsite (N) | 134 | 339 | | 100 | 168 | 83 | 391 | 1215 |
| Immediate follow-up (N) | | 1759 | | 32 | 25 | 9 | 41 | 1866 |
| Six-month follow-up (N) | 229 | 437 | | 17 | 13 | 0 | 14 | 710 |
| Longitudinal | | | | | | | | |
| Pre-post (N) | | 68 | 116 | | | | | 184 |
| Onsite-post (N) | | 42 | | 62 | 22 | 0 | 22 | 148 |

*Note:* An additional sample of 6649 was collected for a small number of questions in the immediate follow-up to Burning Man 2016.

predicting that anticipation would positively predict transformative experience.

Setting may also be important: transformative experiences may be particularly associated with moral expansion when they occur in the context of explicitly communal event norms[51,52]. This question remains unexplored because, with some exceptions[18,27], past research on transformative experiences and mass gatherings has focused largely on singular events. Here, we took a comparative approach, exploiting natural variation across field sites in the salience of communal norms. Specifically, some of our sites promoted communal norms by operating a gift economy that explicitly prohibited the use of money, while other sites promoted individualistic norms by operating a market economy where money could be exchanged for goods and services[53]. If setting matters, prosocial behavior may be more strongly associated with transformative experiences that occur within gifts as opposed to market economies.

Overall, we set out to answer several questions about transformative experiences at secular multi-day mass gatherings, building upon and extending past work on mass gatherings, identity fusion, and prosocial behavior. First, we sought to describe the psychological qualities of transformative experiences as they unfolded over time and document how they relate to expectations and desires for transformation. Second, we examined the prosocial correlates of transformative experiences, investigating their potential to enhance generosity and expand the moral circle by engendering feelings of universal connectedness. Finally, we measured the persistence and evolution of transformative experiences and their prosocial correlates in the weeks and months following mass gathering attendance.

## Results

We collected data at six events across five field sites that varied on several key attributes, including the total number of attendees, location, and the presence of a gift versus market economy (Table 1). Over a data collection period of 5 years, we were able to obtain data from 1215 onsite participants. Gift economy field sites included Burning Man (an event in central Nevada; $n = 473$) and Burning Nest (a "Regional Burn" event in the UK officially associated with the Burning Man organization; $n = 100$). Market-economy sites included Lightning in a Bottle (a festival in California; $n = 391$), Latitude (a festival in the UK; $n = 168$), and Dirty Bird Campout (a festival in California; $n = 83$). See Table 1 for more information on data collection, SOM 3.1 for detailed descriptions of each event, and SOM 3.2 for detailed participant demographics. We collected self-reported frequency of gift-giving and money handling at each event, and confirmed that people gave more gifts ($t(1036) = 12.60$, $P < 0.001$), received more gifts ($t(1082) = 16.30$, $P < 0.001$), and handled less money ($t(576) = 11.80$, $P < 0.001$), at gift economies than market economies.

We supplemented our onsite data with online surveys administered 0–2 weeks before ("pre-event", $n = 600$), 0–4 weeks after ("immediate follow-up", $n = 1866$), and 6 months after ("6-month follow-up", $n = 710$) attendance, which allowed us to track relationships between transformative experience and moral orientation over time (Fig. 1). Links to an online survey were distributed to e-mail addresses collected from onsite participants ("targeted") as well as newsletters sent out to all mass gathering participants ("untargeted"). In addition, we were able to collect a small number of additional variables in a supplementary sample collected through the Burning Man Census, an annual post-event

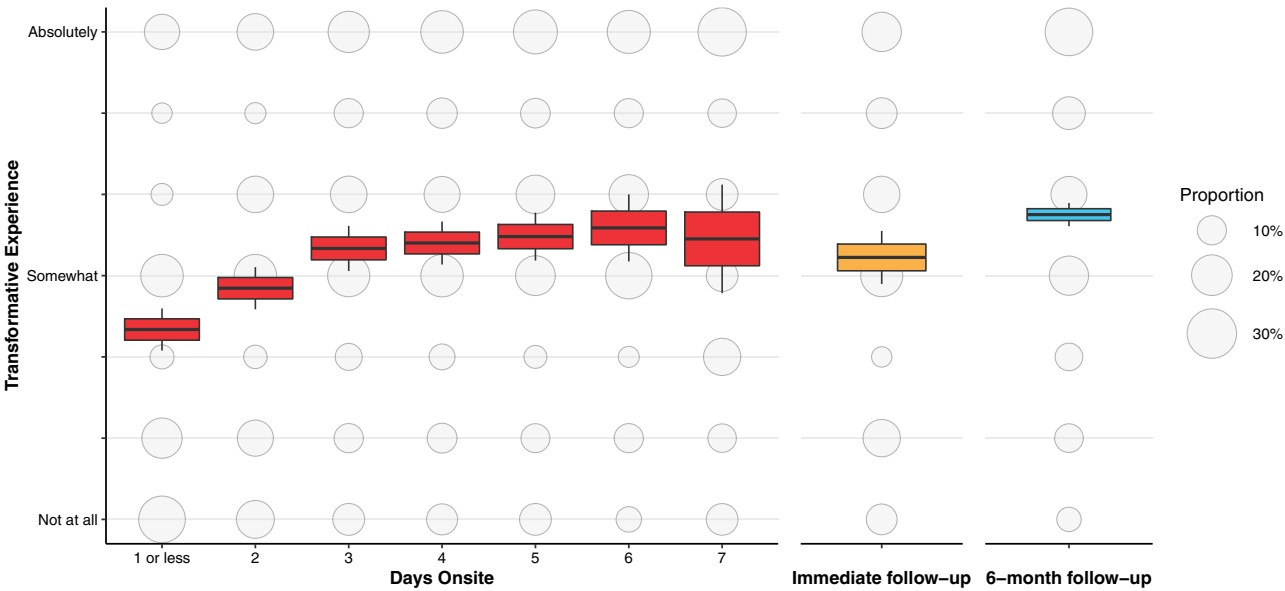

**Fig. 2 Self-reported transformative experience increases over time at mass gatherings and persists at least 6 months after attendance ($n = 3081$ independent samples examined over 3 separate timepoints in 15 separate studies).** Boxplot centers indicate average reported transformative experience at each timepoint; box edges standard error of the mean; whiskers 95% confidence interval. Bubbles reflect the proportion within each timepoint reporting each level of transformative experience. Source data are provided as a Source Data file.

online survey distributed to Burning Man attendees ($n = 6649$). Finally, we collected comparison data ($n = 98$) from a "virtual" mass gathering that took place among Burning Man attendees in the context of the COVID-19 pandemic (see SOM 1.6 for details).

Two within-subjects, repeated-measures (longitudinal) samples were obtained by combining datasets from multiple timepoints using an anonymous identifier code unique to each participant: one that compared responses from the "pre-event" and "immediate follow-up" samples ("pre-post"; $n = 184$), and one that compared responses from the "onsite" and "immediate follow-up" samples ("onsite-post"; $n = 148$). "Cross-sectional" analyses refer to those occurring within a single time period (e.g., onsite, immediate follow-up, etc.); "longitudinal" refer to those that examine relationships between variables collected over time within subjects (Fig. 1). Longitudinal analyses assessed test-retest reliability on all primary measures (transformative experience, universal connectedness, moral expansion, and generosity), all $r$s ≥ 0.40, all $P$s < 0.0001, except that onsite generosity did not significantly correlate with immediate follow-up generosity, $r = 0.15$, $P = 0.23$, possibly because of reduced within-subjects sample size for this measure ($n = 62$; see SOM 1.1 and Supplementary Table S1 for details).

**Transformative experience.** First, we assessed changes in self-reported transformative experience over time onsite. Regression analysis showed that, as predicted, rates of transformative experience increased significantly over time, $B = 0.23$, $SE = 0.03$, $t(1178) = 7.61$, $P < 0.001$; Fig. 2). In order to estimate the average expected rate of transformative experience following mass gathering attendance, we calculated mean self-reported transformation among all onsite participants. Results showed that, overall, 63.2% of participants reported being at least "somewhat" transformed, and 19.5% said they were "absolutely" transformed ($M = 4.08$, $SD = 2.06$). Overall, then, these data show the prevalence of self-reported transformative experience at these events and suggest that rates of transformative experience increase over time.

Next, we examined demographic, affective, and behavioral predictors of transformative experience. In our main models, we did not observe associations between transformative experience and gender, age, or income (all $P$s > 0.3). Transformative experience was negatively associated with educational attainment, $B = -0.12$, $SE = 0.05$, $t(1177) = -2.52$, $P = 0.012$ and the consumption of alcohol, $B = 0.29$, $SE = 0.13$, $t(1177) = -2.24$, $P < 0.025$. Transformative experience was positively associated with mood, $B = 0.33$, $SE = 0.06$, $t(1178) = 5.19$, $P < 0.001$, and the use of psychedelic substances, $B = 0.37$, $SE = 0.13$, $t(1177) = 2.75$, $P = 0.006$. We also built exploratory models to examine the contribution of additional behavioral variables to transformative experiences. These analyses suggested that increased reports of transformative experiences over time could be partially attributed to the formation of new social relationships, gift exchange, and dancing (see SOM 1.13 for details).

To probe the qualities of participants' transformative experiences, we asked them a number of questions (see Fig. 3 and "Methods" for a complete list). The most frequently reported two qualities of participants' experiences were feeling socially connected to something larger than oneself ($M = 4.92$, $SD = 1.88$) and perceiving something new about others ($M = 4.55$, $SD = 1.83$). The least frequently reported qualities were feeling as though one's self had dissolved ($M = 2.81$, $SD = 1.90$) and feeling like a different person than they were before ($M = 3.06$, $SD = 2.00$). Figure 3 displays the relative prevalence and 95% confidence intervals of the base rate of each quality. Time-based analysis suggested that the prevalence of each of these qualities increased significantly over time spent at the event, all $P$s < 0.05, with the exception of feeling spiritually connected to something larger than oneself and expressing one's true self (see SOM 1.8). Overall, this analysis suggests that the most prevalent attributes of transformative experiences were socially oriented (e.g., toward others and the community). In contrast, psychedelic substance use most strongly predicted changes to perceptions of reality and oneself, suggesting that transformative experiences elicited by psychedelics may differ in certain key respects from those arising from mass gathering participation alone (see SOM 1.2 for details).

It is possible that people participate in mass gatherings with strong expectations and desires to be transformed, and these expectations and desires may create self-fulfilling prophecies. To

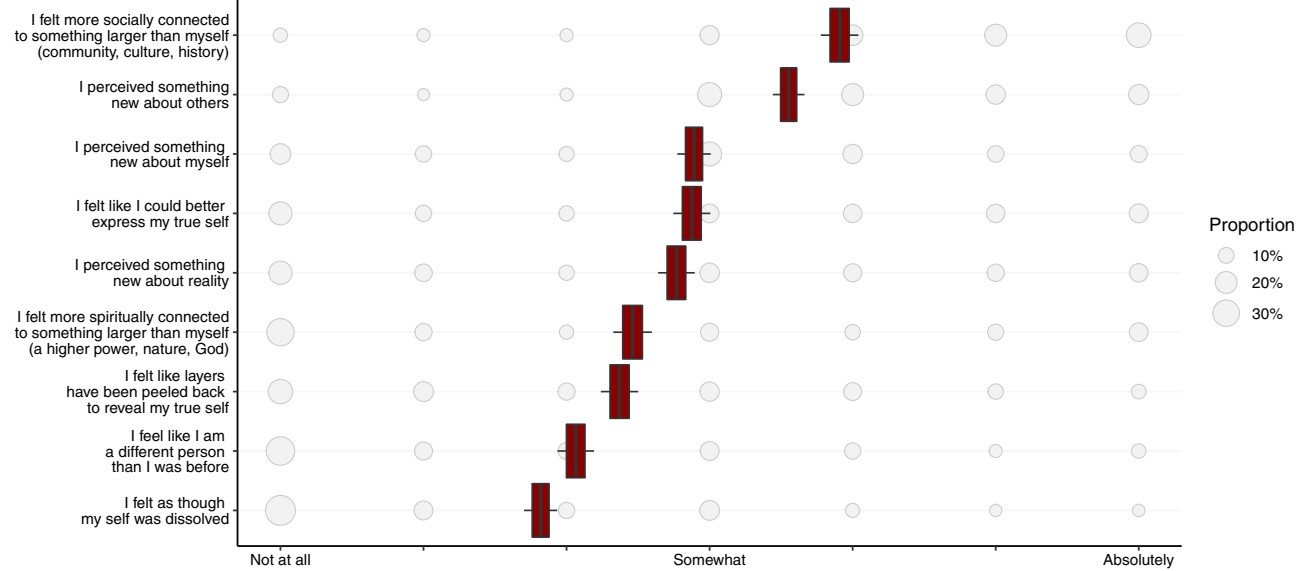

**Fig. 3 Base rates of qualities of transformative experience among onsite participants (n = 1215 observations examined over 6 field studies).** Bubble size reflects proportion providing that response within each question. Box centers indicate the mean value, edges SEM, box whiskers 95% CI. Source data are provided as a Source Data file.

test this question, we tested in a cross-sectional analysis whether expectations and desires for transformation predicted reported transformative experience. Consistent with this notion, both expecting ($B = 0.16$, $SE = 0.04$, $t(1174) = 4.40$, $P < 0.001$) and desiring ($B = 0.23$, $SE = 0.03$, $t(1176) = 6.83$, $P < 0.001$) a transformative experience positively predicted participants' actually having one. It should be noted that, because these measures were collected at the same time as reports on transformative experience, it is possible that transformative experiences onsite subsequently increased reports of expectations and desires for one. Some evidence that speaks against this possibility is that self-reported expectations ($M = 3.19$, $SD = 1.91$) and desires ($M = 3.71$, $SD = 2.09$) collected onsite were, if anything, lower than those collected in the pre-attendance sample ($M = 4.06$, $SD = 1.98$, and $M = 5.38$, $SD = 1.86$; $t(548) = 11.8$, $P < 0.001$ and $t(629) = 14.4$, $P < 0.001$, respectively). We cannot, however, rule out the possibility that transformative experience impacted reports of anticipation.

Notwithstanding the strong relationship between the likelihood of expecting or desiring a transformative experience and the likelihood of having one, there was some evidence the transformative experiences we observed onsite over time were not merely due to the self-fulfilling effects of anticipation. First, we found that, of the 49.6% of participants who did not even somewhat expect to be transformed, nearly half (46.7%) reported being at least somewhat transformed. Similarly, of the 41.5% of people who did not even somewhat desire to be transformed, 42.2% reported being at least somewhat transformed. These results show that considerable proportions of people who neither expected nor desired to be transformed nevertheless reported a transformative experience, thereby supporting the possibility that anticipation is not a necessary condition for transformation. In addition, expectations and desires did not correlate with any of our measures of prosocial attitudes behavior (all $Ps > 0.4$), suggesting that changes in such variables were not the result of anticipation (for additional details regarding the effects of anticipation, see SOM 1.5). Finally, data collected at a "virtual" mass gathering held during the COVID-19 pandemic showed that reports of transformative experience were significantly greater onsite than online, despite greater desires for

transformation in the latter (see SOM 1.6). Overall, these results suggest that while anticipating a transformative experience is positively associated with having one, anticipation is neither a necessary nor sufficient condition for having a transformative experience.

Next, we tested whether reports of transformative experience varied across cultural settings. To do this, we included the cultural context variable (gift versus market economy) and its interaction with time (days onsite) as predictors of self-reported transformative experience, additionally controlling for event location (US versus UK). We then tested the significance of the main effect of setting and its interaction with time on a transformative experience. Results showed no significant effects of cultural context, either as a main effect, $B = -0.10$, $SE = 0.11$, $t(1176) = -0.86$, $P = 0.391$, or as an interaction with time, $B = 0.05$, $SE = 0.06$, $t(1176) = 0.76$, $P = 0.448$. We also examined whether any of the qualities of transformation were more prevalent at gift economies than market economies by examining the main effect of setting on each quality. Results showed that participants of gift economies felt more socially connected to something larger than themselves, $B = 0.38$, $SE = 0.15$, $t(762) = 2.48$, $P = 0.013$. No other effects of cultural setting on transformation qualities were significant. This suggests that the majority of self-reported aspects of mass gathering attendance are consistent regardless of whether or not the event operates a gift economy.

**Prosocial correlates of transformative experience.** Next we sought to test the relationship between the self-reported transformative experience and prosocial orientation. Our analysis focused on two primary dependent variables: generosity and moral expansion. We tested whether these prosocial measures were correlated with self-reported transformative experience and whether they increased over time. In addition, because we hypothesized that changes in prosocial behavior would occur as a result of increased feelings of connectedness to other human beings, we also examined the relationship between these measures and the degree of overlap people reported feeling between themselves and all human beings ("universal connectedness"; see "Methods").

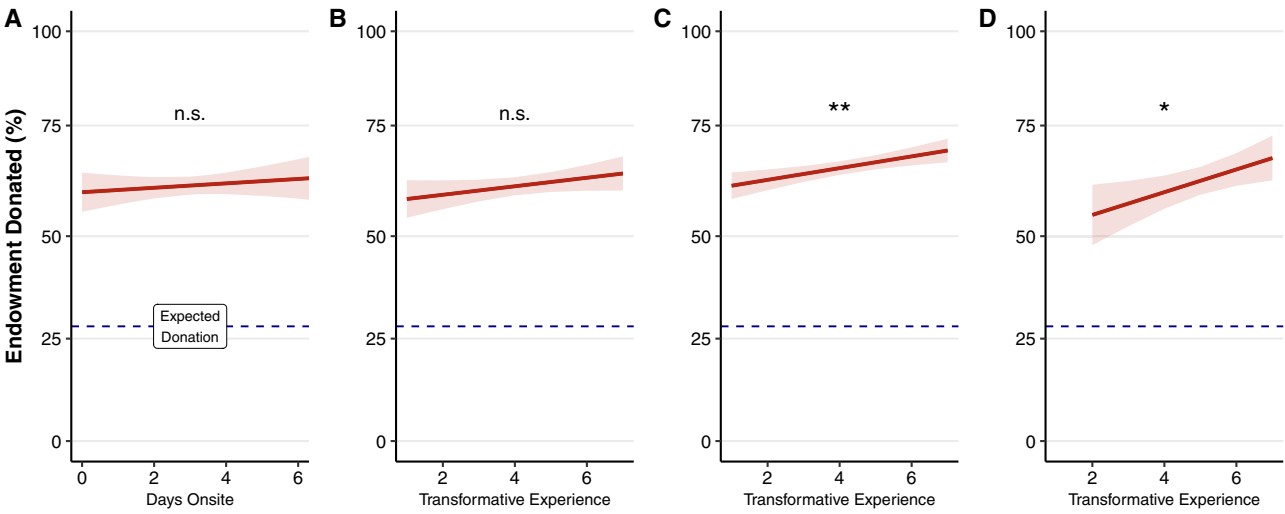

**Fig. 4 Generosity during and after mass gathering attendance. A** The relationship between days onsite and generosity; **B**–**D** The relationship between transformative experience and generosity during ($n = 785$), immediately following ($n = 1810$), and 6 months following ($n = 405$) attendance. Generosity was assessed by providing participants with 10 tickets that would be redeemable for valuable prizes. Participants were given the opportunity to keep as many tickets as they wanted for themselves and give the rest to a randomly selected stranger. Lines show estimated marginal effects using multilevel linear models controlling for incidental variables with event set as a random factor (see analytic approach); significance values based on independently run two-sided $t$-tests on the resulting coefficients. Ribbons indicate 95% confidence intervals. Dashed blue lines reflect the expected level of generosity based on a published meta-analysis of Dictator games 56 ($n = 616$ data points). ns not significant. **$^{**}p = 0.001$; $^*p = 0.012$. Source data are provided as a Source Data file.

To measure generosity at our field sites, we used a modified dictator game[54]. In the classic version of the game, participants are given a monetary endowment then given the opportunity to donate some portion of it to an anonymous stranger. In order to make the game suitable for gift economies (where the use of money is discouraged), we modified the game such that participants were endowed with ten tickets that could each be redeemed for a prize from a "mystery box" containing items of value for eventgoers. Without knowing the exact items in the box, participants decided how many tickets they wanted to give to an anonymous stranger by placing them in an envelope which was then handed to the experimenter. We observed an average donation of 62% of the tickets, a level that is notably higher than donations typically observed in dictator games, which average around 28%[55] (see Fig. 4A). However, levels of generosity did not significantly change over time onsite, $B = 0.05$, $SE = 0.07$, $t(769) = 0.74$, $P = 0.448$, and there was no significant relationship between transformative experience and generosity onsite, $B = 0.11$, $SE = 0.06$, $t(762) = 1.63$, $P = 0.103$ (see Fig. 4B). Furthermore, there was no significant association between universal connectedness and generosity, $B = 0.07$, $SE = 0.07$, $t(762) = 0.97$, $P = 0.334$. Thus, while overall rates of generosity were substantially higher at multiday mass gatherings than typically observed in laboratory studies of dictator games, there was no evidence that generosity was positively associated with transformative experience or with time spent attending mass gatherings.

To measure moral expansion, we used a hypothetical social discounting measure where participants were asked to indicate how much free time they would be willing to spend doing a favor for people at different social distances. We indexed moral expansion by plotting the amount donated for each distance and calculating the area under the curve for each participant[56,57]. Supplementary analyses showed responses on this measure correlated with the classic, hypothetical money-based measure of social discounting, as well as with an incentivized measure of charitable giving across various social distances (see SOM 1.3).

Moral expansion significantly increased with time onsite, $B = 0.07$, $SE = 0.02$, $t(862) = 2.94$, $P = 0.003$, indicating that participants were willing to spend more time helping more socially distant strangers with every passing day spent at mass gatherings (see Fig. 5).

In addition, moral expansion was positively associated with universal connectedness, $B = 0.09$, $SE = 0.03$, $t(960) = 3.43$, $P = 0.001$, which itself positively associated with transformative experience, $B = 0.08$, $SE = 0.03$, $t(1126) = 3.11$, $P = 0.002$. These observations raised the possibility that the effect of time on moral expansion was mediated by its effect on the transformative experience and universal connectedness. To test this hypothesis, we constructed a mediation model testing the significance of the indirect effect from time to moral expansion through transformative experience and universal connectedness. Results showed that a significant indirect effect, $B = 0.002$, $SE = 0.001$, $P = 0.028$, $CI_{95}[0.000, 0.004]$). The direct effect of time on moral expansion remained significant when including this effect in the model, $B = 0.071$, $SE = 0.23$, $P = 0.002$, $CI_{95}[.026, 0.117]$, consistent with partial mediation. A statistical model testing the relationship between time and moral expansion through universal connectedness and transformative experience (as opposed to the reverse, as tested above) showed no significant effects of time on universal connectedness and no significant effects of transformative experience on moral expansion, lending credence to the originally hypothesized variable ordering (see SOM 1.11). Overall, these results suggest that moral expansion increases over time in part as a result of transformative experience and universal connectedness (see Fig. 5).

In supplementary analyses, we explored the possibility that the relationship between time and moral expansion was mediated by group identity fusion—a variable reflecting people's sense of overlap with other event attendees as opposed to other humans in general. First, we substituted group identity fusion for universal connectedness in the model predicting moral expansion from time onsite and transformative experience (see Fig. 4B). Group identity fusion did not significantly predict moral expansion ($B = 0.06$, $SE = 0.04$, $t(809) = 1.50$, $P = 0.135$ versus $B = 0.09$,

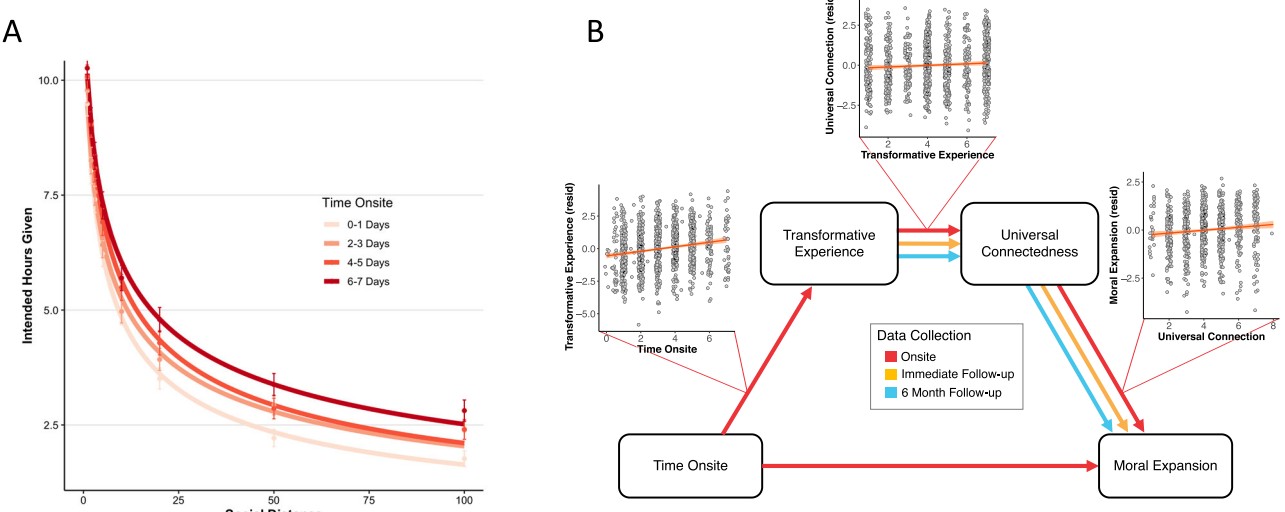

**Fig. 5 Moral expansion at mass gatherings. A** Time spent onsite at mass gathering (in days) was binned into quartiles and average hours intended to give helping others was averaged across level of social distance within each bin. Next, for visualization purposes, a curve was fit to the data of the following function: $y \sim V/(1 + k \times xs)$, where $V$ represents the undiscounted value of the reward (i.e., 14), $y$ the discounted value, and $k$ a parameter representing the degree of social discounting (i.e., the steepness of the curve, see refs). Error bars = SEM. **B** Relationships between time onsite, transformative experience, universal connectedness, and moral expansion onsite; and relationships between the latter three variables in the immediate and 6-month follow-ups. Colored lines represent significant correlations, $P < 0.05$. The scatterplots stemming from each mediation line reflect the predictor variable plotted against the jittered residuals obtained from a regression analysis controlling for all incidental variables. Dot size reflects count; ribbon = SEM. Source data are provided as a Source Data file.

$SE = 0.03$, $t(960) = 3.43$, $P = 0.001$ for universal connectedness), and the indirect effect of time onsite to moral expansion through group identify fusion was not significant, $B = 0.003$, $SE = 0.002$, $P = 0.155$, CI$_{95}$[−0.001, 0.006]. Moreover, a model that allowed both variables to compete for variance by including both as covariates showed that universal connectedness predicted moral expansion even when controlling for group identity fusion ($B = 0.09$, $SE = 0.03$, $t(798) = 3.13$, $P = 0.002$). These results suggest that changes in moral expansion are due more to generalized increases in universal connectedness than to increased connection other eventgoers (see SOM 1.12). It is important to note that universal connectedness and the group identity fusion scales differed in their extremity, with the latter indicating total immersion of the self in the group, while the former indicated only a high degree of overlap, so means of these measures cannot be compared directly. Nevertheless, these results are consistent with the idea that prosocial change at multiday mass gatherings may be the result of a form of universal identity fusion characterized by connectedness to all humanity[36,58,59].

We examined whether the effects of time on generosity and moral expansion varied for events with market versus gift economies by looking at interactions between time and setting on universal connectedness, moral expansion, and generosity. We found no main effects of cultural setting on these measures, nor any interactions between cultural setting and time, all $Ps > 0.05$. Thus we find no evidence that the effect of time on transformative experience and prosocial orientation is moderated by local event norms.

**Persistence of transformative experience and its prosocial correlates**. Next, we examined the relationships between transformative experience, generosity, and moral expansion in the weeks and months following event attendance. Because we considered participants following the event to have received a "full dose" of event attendance (descriptive analysis showed that 99.3% of the sample indicated they stayed for more than 1 day), we

dropped "days onsite" from these analyses and focused on the relationships between transformative experience, generosity, and moral expansion, with universal connectedness as a potential mediator. Because the majority of the follow-up samples consisted of Burning Man attendees, it was not possible to perform cross-event comparisons on these data.

Immediately following attendance ($n = 8515$), 19.5% of participants indicated they had "absolutely" had a transformative experience and 71% said they had at least "somewhat" had a transformative experience. Similar figures were evident six months after attendance ($n = 696$), when 27% of participants reported "absolutely" having a transformative experience and 72.7% reported at least "somewhat" having a transformative experience (Fig. 2). This suggests that the subjective sense of transformation following participation in mass gatherings persists over time.

In our follow-up studies, which were conducted online, participants were endowed with a ten "virtual tickets" redeemable for the chance to win a $50 Amazon gift card. Participants decided the number of tickets they wanted to keep for themselves versus donate to an anonymous stranger. Overall, participants donated 67% of their tickets in the immediate follow-up and 65% of their tickets in the 6-month follow-up.

In contrast to the onsite data, in the immediate follow-up survey, transformative experience was positively associated with generosity, $B = 0.14$, $SE = 0.04$, $t(1801) = 3.26$, $P = 0.001$ (see Fig. 4C). This relationship was replicated in the six-month follow-up, $B = 0.28$, $SE = 0.11$, $t(394) = 2.52$, $P = 0.012$ (see Fig. 4D). To probe these findings further, we examined universal connectedness as a potential mediator. In the immediate follow-up data, the effect of universal connectedness on generosity was not significant, $B = 0.10$, $SE = 0.06$, $t(1762) = 1.60$, $P = 0.111$, nor was the indirect pathway from transformative experience to generosity through universal connectedness, $B = 0.005$, $SE = 0.004$, $P = 0.266$, CI$_{95}$[−0.004, 0.013]. Similar results were observed in the 6-month follow-up data; there was no significant

relationship between universal connectedness and generosity, $B = 0.20$, $SE = 0.13$, $t(394) = 1.51$, $P = 0.131$, and the indirect effect of transformative experience on generosity was not significant, $B = 0.011$, $SE = 0.01$, $P = 0.30$, $CI_{95}[-0.009, 0.03]$. Thus while we observed a relationship between transformative experience and generosity both immediately after event attendance and six months afterwards, we found no evidence for a mediating effect of universal connectedness.

Next, we examined the relationship between transformative experience, universal connectedness, and moral expansion in the weeks and months following event attendance. Immediately following attendance, the relationship between transformative experience and universal connectedness was significant, $B = 0.12$, $SE = 0.02$, $t(1805) = 7.26$, $P < 0.001$, as was the relationship between universal connectedness and moral expansion, $B = 0.11$, $SE = 0.02$, $t(1275) = 4.90$, $P < 0.001$. Mediation analysis showed a significant indirect effect from transformative experience to moral expansion through universal connectedness, $B = 0.023$, $SE = 0.006$, $P < 0.001$, $CI_{95}[0.012, 0.034]$. Similarly, 6 months after attendance, we found a significant relationship between transformative experience on universal connectedness, $B = 0.16$, $SE = 0.04$, $t(397) = 3.86$, $P < 0.001$, a significant effect of universal connectedness on moral expansion, $B = 0.14$, $SE = 0.04$, $t(347) = 3.25$, $P = 0.001$, and a significant indirect effect between these variables, $B = 0.032$, $SE = 0.013$, $P = 0.016$, $CI_{95}[0.006, 0.057]$. This shows that there was a significant indirect relationship between transformative experience and moral expansion through universal connectedness both immediately and 6 months following attendance. In supplementary analyses, we tested a hypothesis that transformative experience-reported onsite would predict moral expansion and universal connectedness following attendance in the same participants, 1–4 months and 6 months later. Although we found some evidence supporting the hypothesis, ultimately the longitudinal sample was not sufficiently powered to permit firm conclusions regarding the long-term effect of transformative experience on prosocial change (SOM 1.4).

## Discussion
Stories of profound personal transformation have long captured the human imagination, yet such experiences are difficult to recreate in the laboratory. Here, we adopted a lab-in-the-field approach to study transformative experiences as they were occurring at several secular multiday mass gatherings in the US and UK. Self-reports of such experiences at these events were common, increased over time, and endured at least six months following attendance. The most prevalent qualities of transformative experience were prosocial in nature and were correlated with increased feelings of connectedness between the self and all human beings. Consistent with these reports, participants showed an expanded moral circle with every passing day, an effect partially mediated by feelings of universal connectedness and transformative experience. Meanwhile, we observed high levels of generosity at mass gatherings, but generosity onsite did not increase over time and was unrelated to the transformative experience. These effects were robust to controlling for expectations and desires for transformative experience as well as substance use, and were consistent across mass gatherings with market economies as well as gift economies. In the weeks and months following event attendance, transformative experience directly predicted generosity and indirectly predicted moral expansion via universal connectedness.

Our results build upon and extend past work on collective effervescence and prosocial behavior, which suggests that mass gatherings played a functional role in human evolution by increasing people's willingness to make personal sacrifices on behalf of the group[10,11,32]. Some research suggests such prosocial behavior is psychologically mediated by experiences of personal transformation[7,14,24] yet thus far research on the prosocial correlates of transformative experiences has mainly relied upon retrospective approaches, which are subject to the limitations of autobiographical memory. Here, in order to better understand how such experiences may be associated with prosocial change, we examined the qualities of transformative experiences as they occurred, and measured their association with prosocial behavior. We found that reports of such experiences did indeed increase over time, and were correlated with an expanded circle of moral regard. This shows not only that such experiences are associated with changes in moral orientation, but also that, in certain contexts at least, such changes may be characterized by feelings of universal moral inclusion[11].

Our findings also complement past research on the psychological consequences of collective rituals showing that such events are associated with a fusing of self and group identities[18,19] and increased group loyalty[17,60]. Much existing research on mass gatherings has focused on cultures that prioritize "binding" values such as in-group conformity and loyalty[6,19,61,62] (though not all such research, see refs. [7–9,28,29]). By contrast, our field sites were located in the US and UK and our samples skewed liberal. Past work shows that these populations prioritize more individualized or impersonal forms of morality (e.g., generosity toward strangers)[62–64]. Thus our results, considered alongside observations from the prior literature, raise the intriguing possibility that transformative experiences at mass gatherings serve to amplify the moral orientation of the local culture. We speculate that in cultures dominated by impersonal or individualized morality, this could manifest as feelings of universal connectedness and moral expansion, whereas in cultures that emphasize binding values, transformative experiences may manifest as group identity fusion and greater generosity toward members of the group.

One possible explanation for our findings is that they represent a giant cultural placebo effect: that the kinds of people who attend secular multiday mass gatherings are especially likely to desire and expect transformative experiences from attending these events, and want their experiences to be socially meaningful. If this is the case, then our results would merely reflect participants' self-reporting they got exactly what they came for. This argument is supported by the fact that moral expansion but not generosity in the dictator game increased over time onsite. While the dictator game involved sharing of real personal resources, the moral expansion measure relied entirely on self-report, leaving open the possibility that participants demonstrated prosocial change only when the personal stakes were low.

We think this explanation is unlikely to account for all our observed effects, for several reasons. First, while generosity did not change onsite, it showed markedly high rates overall—approximately twice as high as average levels of generosity measured in a recent meta-analysis of dictator game studies[55] thus speaking against the view that generosity onsite was merely the result of self-presentational motivations. It is possible that high initial base rates created a ceiling effect from which further increases were statistically difficult to detect, thereby explaining the lack of change in generosity onsite over time. In addition, we observed a positive relationship between transformative experience and generosity in the follow-up surveys conducted 1–4 weeks and 6 months following event attendance. This delayed onset of generosity following transformative experiences is consistent with research showing that highly intense social experiences tend to be followed by periods of reflection in which the personal significance of such events is consolidated into the self-concept[11,41]. In this way, our research extends previous work on

the sacrifices made by individuals on behalf of others following instances of personal transformation and subsequent feelings of identity fusion[10,32]. Further work is necessary to investigate the longer-term trajectories of transformative experience and prosocial change.

Our data on expectations and desires for transformative experience also speak against the possibility that our participants merely reported getting what they came for. First, while expectations and desires did predict the likelihood of having a transformative experience, they were neither necessary nor sufficient conditions for doing so. Even many of those who did not expect or desire to be transformed nevertheless ended up reporting a transformative experience; about one in eight attendees who said they did "not at all" expect to have a transformative experience ended responding that they "absolutely" had one (see SOM 1.5). Furthermore, such expectations and desires did not predict moral expansion or generosity, and we controlled for expectations and desires in all analyses, suggesting that any positive associations reported emerged over and above the effects of anticipation. In addition, examining the temporal dynamics of transformative experiences and their prosocial correlates allowed us to control for potential effects of a general willingness to attend mass gatherings on all of our outcome variables. Because participants completed our study at relatively random timepoints during the mass gathering, we were able to show that reports of transformative experience and moral expansion increased over time at mass gatherings within our study population. Overall, this suggests that these observations are not purely the result of placebo or self-presentation effects.

Our research highlights both parallels and distinctions between the psychological effects of transformative experiences at secular mass gatherings and those triggered by the use of psychedelic substances. We previously reported that psychedelic substance use at mass gatherings predicts experiences of personal transformation, and that such experiences are associated with universal connectedness and positive mood[44]. Yet here we find that transformative experiences at mass gatherings increase over time over and above that expected by psychedelic substance use. In supplementary analyses, we also explored the effects of psychedelics on psychological qualities of transformative experience (see SOM 1.2). While, controlling for substance use, feeling more socially connected to community, culture or history and perceiving something new about others were the most prevalent qualities of transformation reported in the onsite sample, psychedelic substance use was most strongly associated with changes to perceptions of reality and oneself, and least strongly associated with connection to community, culture, or history. This suggests that, while the use of psychedelic substances undoubtedly plays a significant role in eliciting transformative experiences[7], such experiences might be psychologically distinct from those that emerge from mass gathering participation alone.

Our research is subject to a number of important limitations. The most crucial one concerns the generalizability of our findings beyond our study population and field sites. Because most of our participants attended these mass gatherings because they chose to be there, we cannot speak to whether these results would hold for those who do not choose to attend such events, or for events that take place outside the US or UK. In addition, due to the self-selecting nature of the events we studied, our sample is not demographically representative of the country populations in which these events occurred (see SOM 3.2 for further details). More research will be needed to determine the extent to which our observations would apply beyond our study population. It is also possible that, because participants were not randomly assigned to participate, but instead volunteered, that self-selection effects could have led us to over- or under-sample participants

having a transformative experience. Furthermore, because the data collected in the follow-up surveys were cross-sectional (i.e., correlated within timepoint), it is not possible to determine from these data whether transformative experiences are causing prosocial behavior or merely associated with it. We also did not collect data on personality traits, like openness to experience, that might moderate these results in important ways. Investigating how personality traits interact with behavior at mass gatherings to produce transformative experiences will be an important topic for future research.

It is also worth noting some inconsistencies in our research findings. First, while the indirect relationship between transformative experience and moral expansion was mediated by universal connectedness at all three timepoints, universal connectedness did not directly predict generosity at any timepoint. One possible explanation for this is that moral expansion entails a deepening sense of one's interrelatedness with distant others, which is reflected in the universal connectedness measure, while generosity, which was directed toward another anonymous eventgoer, reflected a more localized form of prosociality. Furthermore, while we predicted that time and transformative experience would be associated with increased generosity onsite, no such relationships emerged; instead, transformative experience predicted generosity only in the immediate and six-month follow-up surveys. As noted above, one potential explanation for the delayed onset of this relationship is that personal reflection is required for transformative experiences to take their effect on prosocial behavior[41].

Finally, it is important to note that while our research focuses on transformative experiences at multiday secular mass gatherings, this is by no means the only environment where such experiences can occur. Apart from psychedelic experiences, these include interacting with literature and music[65], practicing meditation[66], or immersing oneself in nature[67]. Given the diversity of these settings, it remains unclear exactly which aspects of mass gathering attendance cause transformative experiences. While our study was not primarily designed to answer this question, our analyses suggest that activities such as dancing (which can elicit emotional synchrony[18,21,29]), giving and receiving gifts (which may amplify prosocial emotions[68], and making new friends (which may increase feelings of mutual obligation[6,8,10]) partially mediated the relationship between time onsite and transformative experience (SOM 1.13 for more details). Other research has identified sleep deprivation and exposure to powerful rhythmic music as additional causes of transformative experience[7]. Additional research will be needed to better understand how these factors interact with aspects of the local event culture and individual differences to produce prosocial transformation.

As our world grows ever more connected, the fates of humans around the globe are becoming increasingly intertwined. Some philosophers have argued that moral progress in the age of globalization will require expanding our circle of moral regard beyond our immediate social group to encompass all of humanity including future generations[69,70], and that pursuing such moral transformation, whether through educational or even biochemical means, is itself morally required[71]. While our findings cannot speak to whether transformative experiences are normatively desirable, they provide evidence for how such experiences at secular multiday mass gatherings persist over time and coincide with changes in moral orientation.

## Methods

**Overview**. Here we summarize our procedures, participants, data collection strategy, and analytic strategy. Variables reported here were those used for primary analyses; see SOM Appendix A for a complete description of all variables collected. The project was approved by the University of Oxford Research Ethics Committee (#MS-IDREC-C1-2015-134).

*Field site selection*. We identified a set of field sites that varied independently on several features of scientific interest (see Table 1). After identifying two gift-economy field sites (Burning Man and Burning Nest), we then sought other locations that matched these events on critical features. Ultimately, we selected two market-economy field sites on the West coast of the United States (Dirty Bird and Lightning in a Bottle) to match the geographical location of Burning Man; we also selected another market-economy event in the UK (Latitude) to match the location of Burning Nest. After we had identified the relevant locations, we reached out to local event organizers to obtain permission to collect data at the events.

*Onsite data collection*. At each of the field sites, 6–8 volunteer research assistants with a background in psychology and ethics training in human subjects research aided in data collection. Data collection was performed in consultation with the third author S. Megan Heller, a cultural anthropologist with experience in field research at mass gatherings. Research assistants were instructed to recruit participants from a booth set up in well-trafficked event areas. Wearing white lab coats, they approached passersby and asked whether they were interested in taking part in an activity called "Play Games for Science". Notably, we did not mention transformative experience or prosocial change in our recruitment efforts in an attempt to minimize selection bias. Prior to data collection, all research assistants attended a training where they were instructed in data collection protocol and practiced the protocol on one another. After providing informed consent, participants answered the survey questions either on paper (Burning Man, Lightning in a Bottle, Latitude) or on electronic tablets (Burning Nest, Dirty Bird). Overall, the study took ~15 min for each participant to complete. Following completion, participants were given an opportunity to collect a prize and thanked for their participation.

*Pre-event data collection*. A biweekly newsletter called Jack Rabbit Speaks is distributed amongst the Burning Man community. We posted an advertisement on this newsletter about the "Psychology of Burning Man" and invited volunteers to participate in a 10–15 min survey about why they go to Burning Man and the effect it has on them. The survey contained all demographic questions, how many times people had previously attended Burning Man, whether people intended to go to this year, whether they were registered and had a ticket to attend, how many other multiday mass gatherings they had attended, whether they expected and desired to have a transformative experience at Burning Man that year. They also completed the universal connectedness measure and the moral expansion measure.

*Immediate follow-up data collection*. Surveys immediately following event attendance were collected in three different ways, from 0 to 4 weeks following the events. First, we sent targeted e-mails to participants of both the pretest and onsite surveys who had provided us their contact information. This survey included the following questions: demographics, previous event attendance, universal connectedness, transformative experience (subjective and epistemic), and moral expansion.

Next, we included a limited number of questions in the Burning Man Census, which is distributed to all attendees. This allowed us to collect data from a large number of participants. Survey questions included on the census were demographics, transformative experience (subjective, epistemic, anticipation), and universal connectedness.

Finally, at the end of the census was a link that gave participants the opportunity to answer more questions about their experience at Burning Man. This "appended" survey included the following additional items: demographics, universal connectedness, and moral expansion.

*Six-month follow-up data collection*. Targeted surveys were sent to onsite participants' e-mail addresses for longitudinal analysis. In addition, we invited readers of Jack Rabbit Speaks, a bimonthly newsletter to the Burning Man community, to take part in the follow-up, which allowed us to collect additional (untargeted) survey data.

*Longitudinal analysis*. In order to be able to identify the same participants across time in longitudinal analyses while still maintaining participant anonymity, we developed a code that would allow them to input the same unique information that would allow them to remain anonymous. Accordingly, in all targeted versions of the survey (pre, onsite, immediate follow-up, and 6-month follow-up), participants were asked to indicate the first three letters of the first road on which they ever lived, the two-digit calendar day of their birth, and the last two letters of their mother's maiden name. In order to match participants across surveys while allowing for the possibility of a typo, we took the Levenshtein distance between the two string variables inserted by participants, and matched participants at 80% similarity, then examined for matches in age and gender. All identifier matches above 80% that also matched in age and gender were considered within-participant subject matches and analyzed longitudinally.

## Participants
*Demographics overview*. Overall, the onsite sample included 625 men, 558 women, 32 fluid/other, with a mean age of 32.4 (*SD* = 11.4) and age range of 17–75. Fifty-seven percent had college degrees, and 37% made over $50,000. The sample skewed liberal, with a mean of 2.6 ("somewhat liberal") on a 7-point scale (1 = "extremely

liberal", 7 = "extremely conservative"). Some differences between the events should be noted. While most events had an average age in the mid-thirties, Lightning in Bottle attendees were significantly younger, with a mean age of 26. This is also reflected in the fact that fewer of them (47%) had graduated college. The event with the lowest income was Burning Nest (16% making $50,000 or more), as contrasted with Burning Man, with an average of 56% at that income level. Overall, participants in the overall sample were not particularly religious, with a mean of 2.3 out of a religiosity scale of 7. The pre-attendance sample (*n* = 600) had a makeup of 242 men, 347 women, 11 = other/fluid, $M_{age}$ = 42.2, *SD* = 13.5. The immediate follow-up sample (*n* = 1866) had a makeup of 962 men, 818 women, 86 = other/fluid, $M_{age}$ = 40.1, *SD* = 12.3. The six-month follow-up sample (*n* = 697) had a makeup of 311 men, 367 women, 19 = other/fluid, $M_{age}$ = 44.1, *SD* = 14.7.

*Exclusions*. Participants were excluded who responded anything other than "0" to the attention check "How many fatal heart attacks have you had?" (*n* = 26). In addition, we excluded participants who reported having been at the event for more than 7 days (*n* = 37, predominantly at Burning Man) since these generally consisted of event organizers, staff, and committed event volunteers.

*Missing data*. In R, general linear models use the listwise deletion method, meaning that single missing value results in the elimination of a participant from the analysis. In order to maximize statistical power in the onsite sample, we estimate missing values using a series of inferential procedures. Missing onsite demographic variables (age (*n* = 9), income (*n* = 13), education (*n* = 9), and religiosity (*n* = 11)), were estimated at event-level means. Missing gender data (*n* = 10) was coded as "other." Missing values of expectations and desires of transformative experience (*n* = 71 and *n* = 73, respectively), consisted mostly of participants (~75%) who had answered "Not at all" on the transformative experience question, so we inferred they had skipped these questions because they had not had a transformative experience. Thus, we estimated missing values of these questions at 1 if the transformative experience was 1, else at event-level means. Due to experimenter oversight, questions regarding transformative experience were not asked in the 6-month follow-up for Lightning in a Bottle; thus, analyses involving this measure at this timepoint were dropped using the listwise deletion method (*n* = 14); due to another oversight, measures of expecting and desiring a transformative experience were not collected in the immediate follow-up of Burning Man 2016; procedures for dealing with this missing data are described in SOM 1.5.2.

## Materials
*Demographics and mood*. Age; gender (1—M; 2—F; 3—Both/Neither/Fluid); mood; education (1-high school to 5-postgraduate degree); income (5 k to over 100 k), politics (1-"extremely liberal" to 7-"extremely conservative"), religiosity (1—not at all religious to 5-very religious). We note that asking about gender but having response options be labeled male and female isn't consistent with current best practices in measuring sex and gender. Since the question asked explicitly about gender, we interpret these data as reflecting gender identity rather than sex. In supplementary analyses, we additionally examine the effects of previous attendance on transformative experiences (SOM 1.9) as well as other potential moderators such as new social connections made and behavioral synchrony (SOM 1.13).

*Days onsite*. To measure the effects of time on prosocial change and transformative experience, we asked participants how many days they had been in attendance. To maximize variance in this measure, we employed a distributed data collection plan in which we collected ~30% of the desired sample on days 0–2, 40% on days 3–4, and 30% on the remaining days. In order to account for different event durations, days onsite were standardized and centered on event-level means. An analysis of daily participant quantities showed this was successful in obtaining significant variance in time onsite in our onsite samples (see SOM 1.2 and Supplementary Fig. S1).

*Substance use*. Participants were reminded that their responses were entirely anonymous and confidential. They were given a list of a variety of substance categories: hallucinogens, euphorics, alcohol, stimulants, narcotics, and cannabis. In each category was at least one legal substance (e.g., salvia, ephedrine, kratom) rendering participants' responses non-self-incriminating. For each substance category, participants indicated whether they were currently under the influence, had taken in the last 24 h, had taken at all that week, and had taken for the 1st time that week. Participants were binary-coded as using psychoactive substances if they selected any of those responses.

*Transformative experience*. We sought for our primary assessment of transformative experience to assess the degree to which participants themselves had had an experience they considered transformative without imposing an external definition. Accordingly, we asked "Have you had a transformative experience at [field site]?" (1—Not at all; 7—Absolutely). To assess the effects of anticipation of transformation, we measured the degree to which participants "expected" and "desired" to have a transformative experience (1—Not at all; 7—Absolutely).

*Qualities of transformation.* We assessed a variety of qualities of transformative experiences via thirteen follow-up questions, full wordings for which can be found in Fig. 3. We also included a question regarding people's "epistemic" transformation which we analyzed in Supplementary Materials (SOM 1.7), as well as the extent, valence, and moral nature of the transformation (SOM 1.10).

*Universal connectedness.* We measured feelings of universal connectedness by adapting the "inclusion of other in the self" measure previously used in relationships research[15]. Typically, participants are presented with a series of seven sets of circles ranging from nonoverlapping to almost entirely overlapping, that represent the "self" and their romantic partner, and asked to indicate which set best describes their relationship. We changed the wording of this question by substituting the romantic partner with "other human beings, in general". This measure was scored on a continuous scale from 1 to 7.

*Group identity fusion.* We employed a measure developed in previous research[10] that shows participants a small circle and a large circle representing the self and the group, respectively. Circles are presented in five consecutive iterations ranging from completely nonoverlapping to the small circle being completely subsumed by the large circle and asked to indicate which set of circles best captured their relationship to the group.

*Moral expansion.* To measure moral expansion, we adapted an established measure of social discounting. In the established version of this measure, participants are asked to list eight individuals ("targets") at different social distances from the self, then asked to indicate how much money they would share with each target. Past work using this measure demonstrates people's tendency to exhibit diminishing generosity toward more socially distant others that typically follows a hyperbolic function. Pilot testing at Burning Man 2015 indicated that the established measure of social discounting was unsuitable for our study population, as participants expressed confusion regarding the use of money-based measures at an event that explicitly prohibited its use. We, therefore, developed a novel social discounting measure suitable for gift economies that asked participants to imagine they had 14 h of free time. They were asked to list the initials of one individual at each of the following social distances from the self: 1, 2, 3, 5, 10, 20, 50, and 100, and then asked them how much time they would be willing to spend doing a personal favor for each of these individuals. In supplementary studies, we confirmed that this time-based measure shows the same hyperbolic discounting pattern as has been observed in the established money-based measure (see SOM 1.3.1 and Supplementary Fig. 4), and that social discounting on the self-report measure used in the current studies correlates with an incentivized measure of social discounting (see SOM 1.3.2 and Supplementary Fig. 5). Following past work[56], we operationalized moral expansion as the area under the curve (AUC) which enables a model-agnostic approach to analyzing discounting data and permits parametric analysis. Participants' degree of moral expansion was computed by plotting each target's social distance against time spent, calculating the resulting AUC, and log-transforming it to normalize it[56].

*Generosity: Onsite.* To measure generosity onsite, we used a modified dictator game[72]. In the classic version of the game, participants are given a monetary endowment and then given the opportunity to donate a portion of it to an anonymous stranger. In order to make the game suitable for gift economies, we initially modified the game such that participants were endowed with five or six tickets that could each be redeemed for a prize from a "mystery box". In reality, the mystery box contained a variety of items that would be of value to eventgoers, including ring pops, snap bracelets, and earplugs (see Supplementary Fig. 2). Without knowing the exact items in the box, participants decided how many tickets they wanted to give to another event participant (chosen at random by the experimenters) by placing the desired number of tickets inside an envelope and handing it to the experimenter. In initial onsite tests of this measure (Burning Man 2015 and Burning Nest, $N = 246$), each ticket was exchangeable for a single item. Data from these samples indicated that participants gave away an average of 72% of their endowment. We were concerned this would limit our ability to detect changes in generosity due to ceiling effects, so for subsequent data collection, we modified the measure to promote more self-interested choices by informing subjects that more desirable items in the mystery box cost more tickets, and increased the number of initially endowed tickets to ten. This was successful, leading to an average donation of 62%. All reported analyses of onsite generosity are based on samples using the modified ten-ticket dictator game.

*Generosity: online follow-ups.* In online follow-up surveys, participants were endowed with a ten "virtual tickets" redeemable for the chance to win a $50 Amazon gift card. Participants decided the number of tickets they wanted to keep for themselves versus donate to another study participant (selected at random by the experimenters).

**Analytic approach.** All onsite data were analyzed using multilevel linear regression models using the *lme4* and *lmerTest* packages in R. Component path analysis (that

is, direct relationships between the predictor, the mediator, and the dependent variables) included field site as a random intercept and controlled for demographics (i.e., gender, age, education, religiosity, and income), and "incidental variables" (i.e., mood, expectations and desires of having a transformative experience, and the use of psychoactive substances [binary-coded as −0.5 or 0.5 and including euphorics, hallucinogens, stimulants, alcohol, narcotics, and cannabis]). Mediation analyses (i.e., tests of indirect relationships between two variables through one or two intermediary variables) were modeled using linear regression via the *lavaan* package in R, controlling for all demographic and incidental variables. All mediation analyses report standardized betas and confidence intervals. Models testing the moderating effects of set (expectations and desires) and setting (gift vs. market economies) controlled for all incidental variables as well as event size and location. Follow-up studies and within-subjects effects were modeled using linear regression adjusting for demographics and incidental variables. See SOM 1.5 for more details regarding effects of expectations and desires.

All data were cleaned and analyzed using R data analysis software with packages *broom 0.7.11, cowplot 1.1.1, tidyverse 1.3.1, forcats 0.5.1, lavaan, 0.6-9, lm.beta 1.5-1, lme4 1.1-27.1, psych 2.1.9,* and *sjPlot 2.8.10.* Analyses examining behavioral and attitudinal effects across events were performed with a mixed model regression using package *lmerTest 3.1-3* in R with event set as random intercept and all other predictors as fixed factors. Mediation analyses were conducted using the *lavaan* package in R (version .5-23.1097). Online surveys were collected in Qualtrics 2015–2020. In all models, parameter estimates were obtained through "normal" maximum likelihood estimation (using the biased sample covariance matrix), with standard errors based on the observed information matrices. Missing values were estimated using a full information maximum likelihood procedure (FIML).

All significant effects are reported in complete regression tables in SOM 5. Small differences between the component path and mediation analyses are due to the fact that the former modeled event-level random effects while the mediation analyses pursued a simple linear approach. Differences in degrees of freedom across statistical tests are the result of missing data or of the fact that certain variables were not assessed at some of the earlier data collection efforts; all instances in which this is the case are noted in the Methods. All other variables and analyses are described in the SOM.

**Reporting summary.** Further information on research design is available in the Nature Research Reporting Summary linked to this article.

## Data availability
The processed data generated in this study have been deposited in the Open Science Framework database at https://osf.io/x5uz9/. Source data are provided in this paper.

## Code availability
All data and scripts, and supplementary materials are available at https://osf.io/x5uz9/.

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

## Acknowledgements

The authors would like to thank BRC Census Lab, Burning Man Organization, the Do Lab, the Crockett Lab, Maher Abdel-Sattar, Valerie Avalos, Helen Bagnall, Dominic Beaulieu-Prevost, Erie Boorman, Kathleen Bryson, Fiery Cushman, Sebastian Deri, Yarrow Dunham, Ross Folkard, Cabe Franklin, Stacy Hackner, Aimie Hope, Kate Hyslop, Katie Joyce, Tobias Kalenscher, Joshua Keay, Vani Kilakkathi, Enoch Lambert, Ashley Lee, Dana Lilienthal Devaul, Theo Maasters-Waage, Tim Muller, David Newman, Cecilia Nunez, L.A. Paul, Matt Plaia, Kelly Peters, Heather Rivers, Judy Saunders, Alexandra Sofrienew, Christopher Timmermann Slater, Daveed Walzer, Caroline Webb, James Whittington, and Kate Wolfe for their research assistance and access to event populations. This research was supported through a grant from the Experience Project from the John Templeton Foundation (ID #49683). The opinions expressed in this publication are those of the authors and do not necessarily reflect the views of the John Templeton Foundation.

## Author contributions

D.A.Y., A.M.B.P., S.M.H., K.M., A.C., and M.C. designed the research; D.A.Y., A.M.B.P., S.M.H., A.C., and M.C. collected the data; D.A.Y. analyzed the data with input from M.C., and D.A.Y. and M.C. wrote the paper, with critical edits from A.M.B.P., S.M.H., K.M., and A.C.

## Competing interests

The authors declare no competing interests.
