## [Peer Review File · Nature Communications]

Prosocial Correlates of Transformative Experience at Secular Multi-Day Mass GatheringsREVIEWER COMMENTS

Reviewer #1 (Remarks to the Author):

The manuscript reports the results of a unique large-scale field study of secular mass gatherings such as Burning Man. I appreciated the large sample sizes, use of multiple timepoints and behavioral measures, and sampling across a variety of mass gatherings with differing internal cultures (market vs. gift economies). The study is novel, well-done, and uniquely able to address new research questions, and the results are important and far-reaching. Thus my overall assessment of this work is very positive. Below I offer a few suggestions for improvement.

1. Although some references are given to support the research questions and hypotheses, I think this needs to be explicated and given more theoretical grounding. Why are these specific relationships (say, between transformative experience and moral expansion) hypothesized, what theories motivate this, and how do the results inform those theories?

2. Similarly, I think some more theoretical justification is needed for the mediation model shown in Figure 5B. Why this ordering of variables and not another?

3. I appreciated the analyses and discussions on the “cultural placebo effect.” I would have liked to see more of this brought in to explain some of the differences between results using direct behavioral measures (e.g., the dictator game) and hypothetical measures (moral expansiveness).

Reviewer #2 (Remarks to the Author):

This was a fascinating paper reporting on a topic that touches base with a number of important theoretical threads; at a general level, how does changes in how we view/experience the self, impact on how we orient ourselves morally towards others. For this reason, I found this work important and interesting. I think the naturalistic setting also adds substantially to the interest of the findings. Overall, the paper was written exceptionally well and the methods, rationale, and data-analysis all seemed to be appropriate.

I did note that the authors had published a subset of these data for a different purpose, but also focusing on transformative experience and social connectedness findings (albeit differently oriented). For this reason, I was especially interested in seeing the evidence for moral expansion. The authors pitched this as an investigation of transformative experiences more generally, but that boiled down to social connectedness and moral expansion (inc. generosity). While there were some other potential moderators explored, these seemed less theoretically relevant.

While they certainly found some very interesting evidence for their main predictions, it was also somewhat less clear than one might have hoped. To summarise, in their onsite data they find that transformative experiences predict moral expansion, but not generosity. In their follow-up data they find that transformative experiences predict generosity, but not moral expansion (on this, I think they have overstated their findings in the abstract, as they say TEs predicted moral expansion at follow-up, but this was only an indirect effect as far as I could tell?). In the on site data, they find that social connectedness mediated the effects of TEs on moral expansion, but were unrelated to generosity. In the follow-up data, they find that social connectedness is unrelated to generosity, but is related to moral expansion – and that there is an indirect effect between TEs and social connectedness on moral expansion. In essence, on these key indicators of moral expansion/pro-social behaviour they observe 2/4 predicted direct effects, and one mediation and one indirect effect with social connectedness.

Given that these are real-world data and likely to be somewhat messy, I think these findings are certainly important and noteworthy. The fact they were able to model time in their data on site is certainly a strength, but as far as I could tell they treated their follow-up data as cross-sectional(?), meaning they could not infer any causality in these data, and this also reduces the level of evidence for these follow-up findings, especially when it comes to the reported indirect effects.

Overall, I think this is a great piece of research and the data are certainly publishable and important – whether they meet the bar for publication in Nature Communications is a question I will leave to the editor.

Reviewer #3 (Remarks to the Author):

The authors found that transformative experiences predict generosity 6 months after an event, but not at the time of the event. They also found that moral expansion occurs via social connectedness (IOS) and transformativeness at mass gatherings. The work is original in its methodology, both the extensive use of natural sites and the adjusted dictator's game. It adds to the literature in these ways but the links between altruism, bonding and transformative experiences have been previously identified. The extension comes regarding the findings on moral expansion. The paper is very well written and clear. The work supports its conclusions well with robust analyses, but they claim that IOS (to humanity) was a stronger measure than identity fusion (to a group of friends), though did not consider the different targets the two measures assessed. The missing piece of the puzzle is what actually causes the transformative experiences at mass gatherings in the first place (possible candidates include synchrony, sleep deprivation, and drugs for example). To that end, drugs are notably very absent from a context where they are, as the authors note, frequently but not ubiquitously consumed. More literature is required and a clearer statement as to why drugs are not analysed here. The methodology is to a high standard, probably reproducible, and meets the standards for field work. The authors could however mention what training (if any) those who collected data in the field had. The control covid online condition does not seem to be a hugely meaningful comparison; it diverges in too many respects, particularly with regard to an absence of synchronised behaviours or relational groups.

Intro

P3

- The empirical research on Durkheimian-type events is not as scarce as the researchers suggest, at least not in cognitive anthropology – there's substantial work by Xygalatas (see firewalking / piercing in Mauritius) or see Newson (on cortisol levels and bonding at mass football events) or Kavanagh on Ju Jitsu belt whipping ceremonies. Newson (2016) and Buhrmester (2018) directly measure personal transformation following intense, dysphoric experiences and how they bond participants together, including subsequent altruism in Buhrmester's paper.

- 'Notably, concern for others beyond one's family and community is a core feature of modern secular morality in Western democratic cultures' – are the authors stating that this is the case specifically for Western democratic cultures and not elsewhere? Or do they refer to these cultures because that's where the study is based? It seems to me that many pre-agricultural societies were/are concerned with life beyond one's community, e.g., nature and a more holistic view of 'us' that incorporates humanity.

P4

- Is the present paper related to the following, if so, why is the latter not mentioned? For instance, the authors state that drug use is controlled throughout to focus on experiences that arose without chemical intervention (surely the reader ought to know that the two studies are connected and indeed that psychedelics also lead to TEs): <https://www.pnas.org/content/117/5/2338.short>

- Likewise, the use of 'set and setting' in the introduction seems unusual in a context that claims to control for drugs...I think it would help the reader to mention that set and setting of drugs is important in lasting outcomes, so too might set and setting of non-chemically induced experiences too.

Fig. 1

- It might be helpful to have red/grey/blue/yellow/grey in the key, rather than (lost) in the legend.

Results

- Is it surprising that people who have been at the event for 4+ days have had more TEs than those there for less time, they've just had more time for them? Assuming this is a between-subject analysis, people who dive right in for a science experiment have less TEs, whereas those who don't bother with

a science experiment the day they arrive may presumably be having more TEs? This could therefore be character/personality driven.

- It's also interesting that TEs go up after 6m – this fits with Whitehouse's work on the need for reflection (Whitehouse and Lanman, 2014) following an event to develop personal transformation (and subsequent group fusion), see e.g., Jong et al., 2016.

P9:

- consider rephrasing 'common intuition' to 'One reason...', common intuition is begging for someone to disagree! For instance, people may participate not because they strongly desire/expect to be transformed, but to reconfirm their previous transformative experiences.

- Under anticipating transformation, could you state whether or not this is within-subject and longitudinal or was this at times a retrospective measure? There are a lot of measures and it will help with the narrative to mention this briefly. If it's retrospective (which it seems to be from the SI), then this is curious – wouldn't the TE (or non TE) influence the ppt's perception of whether they had been expecting it or not? Especially if they hadn't been asked to consider their desires and anticipations and remember them in advance.

- 'Of the 49.6% of participants who did not even somewhat expect to be transformed, 46.7% reported being at least somewhat transformed.' This is purely descriptive – can you put some inferential statistic on this? E.g., chi squared and treat the vars as binary (grouped around the mean?) – those who did not expect to be transformed, I suspect, were significantly no less likely to be transformed than those who did expect to be transformed?

- Same for desires.

P10:

- How does drug intake interact with TEs in relation to desired/anticipated transformation? You mention that you control for drugs, presumably drug consumption is entered as a covariate (I'm reading this article linearly so I expect you'll go on to explain this in methods) – what if people who take drugs do not have their expectations met simply because their experiences go out of their control?

- On P4 of SM, drugs are described as being controlled for in all analyses. Drugs are a big missing step in this paper. The subject matter – TEs that are not drug induced is important and interesting in its own right but (a) why did the authors not just select a site that didn't have drugs at its forefront, e.g., Extinction Rebellion events; and (b) why not also look at the drugs (explaining that there is a sister paper would help with transparency, if this is the case). If the authors explained precisely how drugs were controlled up front, that would help. There is a line that is a little misleading, 'focus on TEs that arose without chemical intervention' – which sounds more like you excluded users of psychoactive substances.

- Drugs are surely a massive variable between the online covid event and the onsite events. Did the online covid event have music/dance or other synchronised activities that generate TEs? Did the online event have sleep deprivation? Did people attending online have relational groups attending with them? As it stands, this control group is not a meaningful comparison. Having just read the SI, it wasn't immediately clear from the main text that the covid event was also a burning man event.

- 'This suggests that the majority of self-reported aspects of mass gathering attendance are consistent regardless of local event norms.' – I disagree, the only local event norm measured was presence of a gift economy so this should be rephrased more specifically. The other festivals may also have more of a gift /trade economy than conventional/mainstream society albeit with the presence of cash. Local event norms at these festivals could also include, for example, freedom of expression, openness to drug taking, creativity etc. So mass gatherings that varied in these facets might produce differing degrees of TEs, but overall the gatherings studied were relatively similar.

Fusion:

- A major problem here is that the targets for the IOS and fusion scales differed so the two cannot be compared (and these targets are not explicitly noted in the body of the text from my reading). Fusion is defined as connection to attendees compared to a larger group. Fusion is not just relationally different but different in its depth, it is a more visceral and enduring bonding – so the final option of fusion (total immersion) is categorically different from connectedness measured by the IOS. This should be stated for readers not familiar with the fusion literature.

- Perhaps it's bonding to humanity that drove moral expansion more than bonding to friends, rather than connectedness over fusion. Certainly, this is what would be suggested by fusion theory (see, for

instance, Swann et al., 2009; 2012 or Whitehouse 2013 – three wishes for the world).

- What happens when social connectedness is subbed out for fusion in the P12 analyses?

P12:

- precisely how many had received a 'full dose' of a gathering, i.e., what % attended for more than a day (there are always dropouts).

- What might explain the relationship between TEs and generosity over time. I'd urge the authors to reconsider the role of identity fusion. Work by Whitehouse (2017; 2018) and colleagues suggests that fusion takes time to emerge, via a process of reflection (Jong et al., 2015) and personal transformation (Newson et al., 2016)...this might explain why prosocial behaviour's relationship with transformation did not immediately appear, rather it appeared once the group was internalised as part of one's core identity via identity fusion.

- The authors seem to point toward global implications and the value of TEs, which brings the paper back to the start. What actually causes a TE???

Methods

- There are a few typos in methods, e.g., 'measures' / 'in category was at least one'

References

The Whitehouse & Lanman (2014) reference in Google Scholar is incorrect, as far as I know and only these two authors were on the original paper.

Reporting summary

- It is very surprising that ppts scored so low for liberalism at what I would perceive to be pretty liberal festivals!

- What exactly were the 'prizes' offered to ppts?

SI

- The authors report several additional online studies in the SI. Presuming these studies are not published elsewhere, these are studies worthy of more credit and could perhaps be mentioned in the abstract ('a series of on site and X online studies').

- However, the SI is extensive. Take SI 1.4 – this reports a whole study with a discussion. Its connection to the main text is not entirely clear, it seems like it needs a space of its own. Was this study integral to the design of the main study, or a failed study that was still worthy of being written up?

- Could the authors condense the text into tables perhaps? E.g. giving %s to Likert type scales in written form is lengthy and a table might be quicker for the reader to extract info from.

- P28 seems to have referencing errors.

Note: reviewer responses (in full) are in bold; our responses are in normal typeface, excerpts in italic. References appear as endnotes at the end of this letter.

Reviewer #1

The manuscript reports the results of a unique large-scale field study of secular mass gatherings such as Burning Man. I appreciated the large sample sizes, use of multiple timepoints and behavioral measures, and sampling across a variety of mass gatherings with differing internal cultures (market vs. gift economies). The study is novel, well-done, and uniquely able to address new research questions, and the results are important and far-reaching. Thus my overall assessment of this work is very positive.

Thank you very much for this assessment!

Below I offer a few suggestions for improvement.

1. Although some references are given to support the research questions and hypotheses, I think this needs to be explicated and given more theoretical grounding. Why are these specific relationships (say, between transformative experience and moral expansion) hypothesized, what theories motivate this, and how do the results inform those theories?

Thank you for encouraging us to provide additional theoretical grounding for our empirical approach. We have now added additional text in the Introduction and Discussion that provide further explanation for our theoretical approach and the implications of our results. Specifically, we write (p. 3):

Testimonial accounts from attendees of mass gatherings suggest that such self-transcendent experiences may be epistemically and personally transformative, resulting in lasting changes to the self^{1,2}. The philosopher L.A. Paul articulates two key components of transformative experiences: they provide new knowledge that is impossible to attain without having the experience, and produce changes in personal values and priorities that cannot be anticipated³. Here, we investigate the psychological qualities of transformative experiences at secular mass gatherings and test the possibility that such experiences are associated with enduring changes in moral orientation.

Support for our hypothesis comes from research showing that collective gatherings such as rituals⁴, ceremonies⁵, raves⁶, and sporting events^{7,8} are associated with feelings of self-expansion and “group identity fusion”^{9,10,11,12}—a psychological state characterized by intense feelings of merging or oneness between the self and the group^{13,14}. This phenomenon is well-documented across cultures¹⁵ and is associated with endorsement of group values^{16,17}, enhanced group loyalty^{18,19}, increased generosity^{20,21,22,23,24,25,26}, and cooperation^{27,28,29,30}. Such prosocial changes are thought to emerge from psychological processes whose original adaptive function lay in promoting the survival of the group at the expense of the self³¹. For example, anthropological accounts suggest that humans throughout history have often engaged in ritualistic behaviors (e.g. ceremonial dancing) that amplify identity fusion before performing acts of extreme self-sacrifice³².

Past work on mass gatherings and identity fusion has tended to focus on how such experiences can amplify prosocial feelings and behavior directed toward the members of one's own social group. Here, by contrast, we investigated the potential for transformative experiences at mass gatherings to expand the boundaries of one's moral circle beyond the group to include all of humanity^{33,34,35}. Recent research suggests that experiences of self-transcendent emotions are associated with increased "identification with all humanity"—i.e., increased concern for all other human beings—and motivations to help distant others^{36,37}. Other work has shown that such experiences are associated with heightened feelings social connectedness³⁸. Building on this and other recent theoretical work³, we conceptualized transformative experiences at mass gatherings as precipitating events that may lead to changes in values and behavior characterized by increased sense of connection other human beings and an expanded moral circle. Thus, we tested whether self-reported transformative experiences at secular mass gatherings were accompanied by increased feelings of universal connectedness, and investigated whether such experiences would be associated with enduring changes in generosity and moral expansion.

In the Discussion, we include the following text that further details how we believe our work informs existing theory on transformative experience and moral expansion (p. 17):

Our results build upon and extend past work on collective effervescence and prosocial behavior, which suggests that mass gatherings played a functional role in human evolution by increasing people's willingness to make personal sacrifices on behalf of the group^{9,10,32}. Some research suggests such prosocial behavior is psychologically mediated by experiences of personal transformation^{6,13,24} yet thus far research on the prosocial correlates of transformative experiences has mainly relied upon retrospective approaches, which are subject to the limitations of autobiographical memory. Here, in order to better understand how such experiences may be associated with prosocial change, we examined the qualities of transformative experiences as they occurred, and measured their association with prosocial behavior. We found that reports of such experiences did indeed increase over time, and were correlated with an expanded circle of moral regard. This shows not only that such experiences are associated with changes in moral orientation, but also that, in certain contexts at least, such changes may be characterized by feelings of universal moral inclusion¹⁰.

Our findings also complement past research on the psychological consequences of collective rituals showing that such events are associated with a fusing of self and group identities^{17,19} and increased group loyalty^{16,39}. Much existing research on mass gatherings has focused on cultures that prioritize "binding" values such as in-group conformity and loyalty^{18,19,40,41} (though not all such research, see refs^{28,29,6,7,8}). By contrast, our field sites were located in the US and UK and our samples skewed liberal. Past work shows that these populations prioritize more individualized or "impersonal" forms of morality (e.g., generosity toward strangers)^{41,42,43}. Thus our results, considered alongside observations from the prior literature, raise the intriguing possibility that transformative experiences at mass gatherings serve to amplify the moral orientation of the local culture. We speculate that in cultures dominated by impersonal or individualized morality, this could manifest as feelings of universal connectedness and

moral expansion, whereas in cultures that emphasize binding values, transformative experiences may manifest as group identity fusion and greater generosity toward members of the group.

2. Similarly, I think some more theoretical justification is needed for the mediation model shown in Figure 5B. Why this ordering of variables and not another?

Following research by Paul (2014) we are conceptualizing transformative experiences as epistemic discoveries that precipitate shifts in one's preferences, values, or beliefs. In other words, the experience generates new knowledge that could not have been anticipated or simulated before the experience, and this new knowledge can have profound subsequent impacts on values and behavior. In the case of transformative experiences at mass gatherings, we hypothesized that transformative experiences precipitate changes in moral orientation such as increased interpersonal connectedness, generosity, and moral expansion. Accordingly, we built our mediation models such that transformative experience precedes universal connectedness and moral expansion, rather than the other way around.

Consistent with this perspective, reversing the ordering of the variables in the mediation model depicted in Figure 5B results in a decrease in model fit. Specifically, Path 1 (from time to universal connectedness) is no longer significant, $B = -0.02$, $SE = 0.03$, $t(1140) = -0.61$, $p = 0.539$, nor is the direct path from transformative experience to moral expansion, $B = 0.01$, $SE = 0.02$, $t(859) = 0.33$, $p = 0.741$.

Prompted by your comment, we have made two changes to the text. First, we have clarified our hypotheses regarding the nature of transformative experience and its relationship to prosocial change. Specifically, we now write in the introduction (p. 3):

we conceptualized transformative experiences at mass gatherings as precipitating events that may lead to changes in values and behavior characterized by an increased sense of connection to other human beings and an expanded moral circle. Thus, we tested whether self-reported transformative experiences at secular mass gatherings were accompanied by increased feelings of universal connectedness, and investigated whether such experiences would be associated with enduring changes in generosity and moral expansion.

Furthermore, we have now included the full reverse-variable model in the SOM 1.11 and referred to it for readers' convenience in the main text as follows (p. 12):

A statistical model testing the relationship between time and moral expansion through universal connectedness and transformative experience (as opposed to the reverse, as tested above) showed no significant effects of time on universal connectedness and no significant effects of transformative experience on moral expansion, lending credence to the originally hypothesized variable ordering (see SOM 1.11).

In SOM 1.11 we write (p. 31):

Following past research⁴⁴, we conceptualized transformative experiences as epistemic discoveries that precipitate shifts in one's preferences, values, or beliefs. In other words, such experiences generate new knowledge that could not have been anticipated before the experience, and this new knowledge is predicted to subsequently impact values and behavior. In the case of transformative experiences at mass gatherings, we hypothesized that transformative experiences precipitate increased feelings of universal connectedness, which in turn predicts increases in moral expansion. Accordingly, we built our mediation models such that transformative experience precedes universal connectedness and moral expansion, rather than the other way around. Thus, in our analyses (see e.g., Figure 3), we modeled transformative experience as being directly associated with time onsite and predicting universal connectedness and moral expansion. On the other hand, it is possible that time onsite leads to increased universal connectedness, which subsequently leads to feelings of personal transformation.

To test this question, we constructed and tested an exploratory mediation model in which time onsite predicted universal connectedness, which subsequently predicted transformative experience and moral expansion. All paths showed significantly reduced model fit than the original model specification. The total indirect effect was no longer significant, $B = 0.00$, $SE = 0.00$, $p = 0.738$, $CI_{95}[0.00, 0.00]$, nor was the effect of time on universal connectedness, $B = -0.02$, $SE = 0.03$, $t(1140) = -0.61$, $p = 0.539$, nor was the effect of transformative experience on moral expansion, $B = 0.01$, $SE = 0.02$, $t(859) = 0.33$, $p = 0.741$. Thus reversing the variable order here does not appear to provide a more robust explanation for the relationships between the variables.

3. I appreciated the analyses and discussions on the “cultural placebo effect.” I would have liked to see more of this brought in to explain some of the differences between results using direct behavioral measures (e.g., the dictator game) and hypothetical measures (moral expansiveness).

We have now added text in the Discussion giving a more detailed description of this possibility. In particular, we have now written (p. 17):

One possible explanation for our findings is that they represent a giant cultural placebo effect: that the kinds of people who attend secular multi-day mass gatherings are especially likely to desire and expect transformative experiences from attending these events, and want their experiences to be socially meaningful. If this is the case, then our results would merely reflect participants self-reporting they got exactly what they came for. This argument is supported by the fact that moral expansion but not generosity in the dictator game increased over time onsite. While the dictator game involved sharing of real personal resources, the moral expansion measure relied entirely on self-report, leaving open the possibility that participants demonstrated prosocial change only when the personal stakes were low.

Reviewer #2

This was a fascinating paper reporting on a topic that touches base with a number of important theoretical threads; at a general level, how does changes in how we

view/experience the self, impact on how we orient ourselves morally towards others. For this reason, I found this work important and interesting. I think the naturalistic setting also adds substantially to the interest of the findings. Overall, the paper was written exceptionally well and the methods, rationale, and data-analysis all seemed to be appropriate.

Thank you for such a positive assessment!

I did note that the authors had published a subset of these data for a different purpose, but also focusing on transformative experience and social connectedness findings (albeit differently oriented). For this reason, I was especially interested in seeing the evidence for moral expansion. The authors pitched this as an investigation of transformative experiences more generally, but that boiled down to social connectedness and moral expansion (inc. generosity). While there were some other potential moderators explored, these seemed less theoretically relevant.

Thank you for bringing this to our attention; we mistakenly omitted the citation (Forstmann et al., 2020) for the other paper that (as you correctly observe) analyzed a subset of the onsite data to explore a different theoretical question: how consumption of psychoactive substances – particularly psychedelics – impacts well-being. We apologize for the oversight and have rectified it in the current version. We agree that the most novel aspects of the current submission concern the relationship between transformative experience and generosity and moral expansion. We now clarify the relationship between Forstmann et al. (2020) and the present work on p. 4:

We designed a variety of experimental procedures and methods appropriate for the mass gathering context, including measures of generosity and moral expansion that did not rely on the use of money, which was prohibited at some of our field sites (see SOM 4 for full descriptions of measures and verbatim participant instructions). We also collected detailed measures of participants' use of psychoactive substances, as past work has implicated psychedelic substance use in transformative experience and prosocial behavior^{6,45,46}. In related work, we have examined the emotional consequences of psychedelic substance use at mass gatherings⁴⁷, finding that their use is associated with increased positive mood—an effect mediated by transformative experience and universal connectedness. Here, we sought to examine the nature and prosocial correlates of transformative experiences that do not necessarily arise from psychedelic substance use. To this end, we controlled for substance use in all analyses of onsite data, and compared the psychological qualities of transformative experience that arose in the presence versus absence of psychedelic substances (see SOM 1.2 for further details).

While they certainly found some very interesting evidence for their main predictions, it was also somewhat less clear than one might have hoped. To summarise, in their onsite data they find that transformative experiences predict moral expansion, but not generosity. In their follow-up data they find that transformative experiences predict generosity, but not moral expansion (on this, I think they have overstated their findings in the abstract, as they say TEs predicted moral expansion at follow-up, but this was only an indirect effect as far as I could tell?). In the on site data, they find that social connectedness mediated the effects

of TEs on moral expansion, but were unrelated to generosity. In the follow-up data, they find that social connectedness is unrelated to generosity, but is related to moral expansion – and that there is an indirect effect between TEs and social connectedness on moral expansion. In essence, on these key indicators of moral expansion/pro-social behaviour they observe 2/4 predicted direct effects, and one mediation and one indirect effect with social connectedness.

Given that these are real-world data and likely to be somewhat messy, I think these findings are certainly important and noteworthy. The fact they were able to model time in their data on site is certainly a strength, but as far as I could tell they treated their follow-up data as cross-sectional(?), meaning they could not infer any causality in these data, and this also reduces the level of evidence for these follow-up findings, especially when it comes to the reported indirect effects.

Overall, I think this is a great piece of research and the data are certainly publishable and important – whether they meet the bar for publication in Nature Communications is a question I will leave to the editor.

Thank you for providing such a detailed summary of many of our observations. We acknowledge that our results do not entirely fit into a “clean” narrative, which can be expected from real-world data. However, despite the inconsistencies you point out, our main finding – the indirect effect of transformative experience on moral expansion via universal connectedness – is seen at all three time points.

In considering how to describe these findings, we decided to report the data transparently and allow readers to decide for themselves what conclusions to draw from them, as opposed to presenting only the data that fit into a tidy narrative. To further improve transparency, we have expanded our discussion of the limitations of the data, in particular making note of the inconsistencies that you refer to in your comment. Specifically, we write in the Discussion (p. 19):

*It is also worth noting some inconsistencies in our research findings. First, while the indirect relationship between transformative experience and moral expansion was mediated by universal connectedness at all three time points, universal connectedness did not directly predict generosity at any timepoint. One possible explanation for this is that moral expansion entails a deepening sense of one’s interrelatedness with distant others, which is reflected in the universal connectedness measure, while generosity, which was directed toward another anonymous eventgoer, reflected a more localized form of prosociality. Furthermore, while we predicted that time and transformative experience would be associated with increased generosity onsite, no such relationships emerged; instead, transformative experience predicted generosity only in the immediate and six-month followup surveys. As noted above, one potential explanation for the delayed onset of this relationship is that personal reflection is required for transformative experiences to take their effect on prosocial behavior***Error! Bookmark not defined..**

We also write (p. 19):

because the data collected in the followup surveys was cross-sectional (i.e., correlated within timepoint), it is not possible to determine from these data whether transformative experiences are causing prosocial behavior or merely associated with it.

We are also grateful to you for pointing out that our abstract implied that transformative experience directly predicted moral expansion in the followup samples, when in fact it did so only indirectly via universal connectedness. We have now changed the text accordingly (p. 2):

Immediately and six months following mass gathering attendance, reports of transformative experience were persistent, directly predicted generosity, and indirectly predicted moral expansion via universal connectedness.

Reviewer #3 (Remarks to the Author):

The authors found that transformative experiences predict generosity 6 months after an event, but not at the time of the event. They also found that moral expansion occurs via social connectedness (IOS) and transformativeness at mass gatherings. The work is original in its methodology, both the extensive use of natural sites and the adjusted dictator's game. It adds to the literature in these ways but the links between altruism, bonding and transformative experiences have been previously identified.

The extension comes regarding the findings on moral expansion.

The paper is very well written and clear. The work supports its conclusions well with robust analyses, but they claim that IOS (to humanity) was a stronger measure than identity fusion (to a group of friends), though did not consider the different targets the two measures assessed.

Thank you for this positive assessment! We agree that we did not adequately describe the identity fusion findings, and have made a number of changes to rectify this (see below).

The missing piece of the puzzle is what actually causes the transformative experiences at mass gatherings in the first place (possible candidates include synchrony, sleep deprivation, and drugs for example). To that end, drugs are notably very absent from a context where they are, as the authors note, frequently but not ubiquitously consumed. More literature is required and a clearer statement as to why drugs are not analysed here.

Thank you for bringing up this important point. Due to an oversight, we inadvertently omitted a citation (Forstmann et al., 2020) that provides context for how the current paper relates to the prior research on the use of psychoactive substances at mass gatherings. We have rectified the error (see p. 4). In addition, inspired by your comments, we have now provided a number of additional clarifications and analyses throughout the main paper and the supplementary materials regarding drug use. For example, in the Introduction (p. 4), we write:

We designed a variety of experimental procedures and methods appropriate for the mass gathering context, including measures of generosity and moral expansion that did not rely on the use of money, which was prohibited at some of our field sites (see SOM 4 for full descriptions of measures and verbatim participant instructions). We also collected detailed measures of participants' use of psychoactive substances, as past work has implicated psychedelic substance use in transformative experience and prosocial behavior^{6,48,49}. In related work, we have examined the emotional consequences of psychedelic substance use at mass gatherings⁵⁰, finding that their use is associated with increased positive mood—an effect mediated by transformative experience and universal connectedness. Here, we sought to examine the nature and prosocial correlates of transformative experiences that do not necessarily arise from psychedelic substance use. To this end, we controlled for substance use in all analyses of onsite data, and compared the psychological qualities of transformative experience that arose in the presence versus absence of psychedelic substances (see SOM 1.2 for further details).

Additionally, we now include a full section in the SOM on the effects of drug use (in particular, the use of psychedelic substances) on prosocial outcomes and types of transformation, none of which has been previously reported (SOM 1.2). We direct readers to these new analyses in the Results section on p. 10, highlighting that psychedelic use predicts qualitatively distinctive transformative experiences relative to mass gathering attendance alone:

Notably, psychedelic substance use most strongly predicted changes to perceptions of reality and oneself, suggesting that transformative experiences elicited by psychedelics may differ in certain key respects from those arising from mass gathering participation alone (see SOM 1.2 for details).

For your convenience, we have pasted SOM 1.2 below, which can also be found on p. 4 of the SOM.

1.2. Drug use.

Common perception of many of the multi-day mass gatherings we surveyed is that they are a site of prevalent psychoactive substance use. Therefore, it was important to determine the degree to which the use of such substances was related to, or even responsible for, the outcomes we observed. Elsewhere we have reported the mood-enhancing effects of psychedelic substance use and its relationship to transformative experience and universal connectedness⁵¹. We find that psychedelic use is associated with positive affect, an effect sequentially mediated by transformative experience and universal connectedness. Here we test how psychedelic substance use related to anticipating a transformative experience, as well as the prosocial outcomes tested (i.e., moral expansion and generosity).

1.2.1. Prevalence. *Table S2 shows the overall use of psychoactive substances onsite. To determine the number of people who did not use any psychoactive substances, we computed two values: first, the percent of participants who said they used no substances whatsoever, and second, the percent of participants who said they used no substances controlled by the US government (i.e., no substances except alcohol).*

Table S2. Frequency of substance use in onsite sample.

Substance	% reported using onsite
Alcohol	81
Cannabis	52
Psychedelics	28
Euphorics	25
Stimulants	22
Narcotics	3
No controlled	39
None	13

1.2.2. Psychedelics and anticipation. *First, we tested the relationship between drug use and anticipation of transformative experience (i.e., expectations and desires). This was based off the possibility that drug use might have changed people's expectations or caused their*

experiences to go out of their control. Based off prior research showing that psychedelics (but not other substances) predicted transformative experiences at mass gatherings, and in order to avoid inflated alpha levels stemming from multiple significance tests, we focused our tests on the use of psychedelics (hallucinogens). As specified in the main paper, we entered a model that specified self-reported use of psychedelic substances onsite as either -.5 or .5, then tested whether it predicted desires and expectations for transformative experience, controlling for all incidental variables. Results showed that people who used psychedelics reported both greater desires, $B = 0.36$, $SE = 0.14$, $t(1192) = 2.61$, $p = 0.009$, and greater expectations, $B = 0.35$, $SE = 0.13$, $t(1179) = 2.73$, $p = 0.006$, of transformation. This shows that people who used psychedelics were more likely to both expect and desire to have a transformative experience.

Next we tested whether psychedelic substances interacted with expectations and desires to predict self-reported transformative experience. As previously reported (Forstmann et al., 2020), there was a main effect of psychedelic substance use on transformative experience, $B = 0.44$, $SE = 0.12$, $t(1183) = 3.60$, $p < 0.001$. However, no interactions with expectations or desires emerged, both $ps > .5$.

1.2.3. Psychedelics and qualities of transformation. We next explored whether psychedelic substance use was more strongly associated with certain types of transformative experience than others. To do this, we conducted a series of linear regressions controlling for incidental variables with each transformation quality as the dependent measure and extracted the beta weight of psychedelic substance use, which reflected the additional degree of each transformation that could be expected for someone who had used psychedelics onsite. Confidence intervals were Bonferroni-corrected to account for multiple tests. Results (seen in Figure S2 below) showed that, with the exception of feeling socially connected to something larger than oneself including “community, culture, history,” psychedelic-substance users showed an increased likelihood of reporting each quality of transformative experience (all $ps < .01$). In particular, psychedelic-substance users were more likely to say they perceived something new about reality or about themselves, and felt more spiritually connected to “a higher power, nature, God.”

Figure S2. The predicted increase of each quality of transformative experience among onsite users of psychedelic substances (approximately 28% of sample). Results reflect beta coefficient obtained from a linear model with each transformation type as the dependent variable and the binary-coded psychedelic substance use variable as the predictor, controlling for all incidental variables. Error bars indicate Bonferroni-corrected 95% confidence intervals.

1.2.4. Psychedelics and prosocial outcomes. Next, we examined the drug use \times anticipation interaction on universal connectedness, moral expansion, generosity, respectively. As previously reported, there was a main effect of psychedelic substance use on universal connectedness, $B = 0.36$, $SE = 0.11$, $t(1128) = 3.37$, $p = 0.001$. However, there were no main effects or interactions between psychedelic substance use and moral expansion or generosity, all $ps > .2$. Finally, we tested a model examining the interaction of transformative experience and psychedelic substance use on prosocial outcomes. No interactions emerged, all $ps > 2$. Overall, this analysis suggests that, while psychedelic substance use is strongly predictive of self-reported transformative experiences and experiences of universal connectedness, it does not appear to predict shifts in moral expansion or generosity.

Regarding the question of what *causes* transformative experience, please see our response on pp. 26-28 of this letter.

The methodology is to a high standard, probably reproducible, and meets the standards for field work. The authors could however mention what training (if any) those who collected data in the field had.

We now include the following additional information in our Methods section (p. 20):

At each of the field sites, 6-8 volunteer research assistants with a background in psychology and ethics training in human subjects research aided in data collection. Data collection was performed in consultation with third author S. Megan Heller, a cultural anthropologist with experience in field research at mass gatherings. Research assistants were instructed to recruit participants from a booth set up in well-trafficked event areas. Wearing white lab coats, they approached passersby and asked whether they were interested in taking part in an activity called “Play Games for Science.” Notably, we did not mention transformative experience or prosocial change in our recruitment efforts in an attempt to minimize selection bias. Prior to data collection, all research assistants attended a training where they were instructed in data collection protocol and practiced the protocol on one another.

The control covid online condition does not seem to be a hugely meaningful comparison; it diverges in too many respects, particularly with regard to an absence of synchronised behaviours or relational groups.

We agree that there are numerous ways that the Covid online control deviates from the onsite conditions. The purpose of this experiment was to hold one variable constant—the population of attendees—which allowed us to test whether this particular population of eventgoers is predisposed to report transformative experiences, regardless the type of event they are attending.

We wish to describe this study, which, despite its flaws, we think has *some* informational value, as transparently as possible. To that end, we’ve now included a fuller discussion of the study’s limitations in the SOM Discussion (SOM 1.6.4; p. 26):

***1.6.4. Limitations.** The purpose of this study was to assess differences in the level of transformative experiences reported among Burning Man attendees at a virtual versus in-person event. Our results did show such differences, suggesting that characteristics of the sample population are not solely responsible for the high rates of transformative experience reported onsite. At the same time, there are a large number of other factors that varied between the groups that limit the conclusions that can be drawn from this study, including drug use, behavioral synchrony (or any physical motion), sleep deprivation, physical hardship, physical contact with others, dancing, singing, shared food and rituals around eating and food preparation, etc. Thus it is impossible to specify what, exactly may have led to the greater prevalence of transformative experiences at the onsite gathering. Further research will be needed to better understand the psychological differences between virtual and in-person events.*

Intro

P3

- The empirical research on Durkheimian-type events is not as scarce as the researchers suggest, at least not in cognitive anthropology – there’s substantial work by Xygalatas (see firewalking / piercing in Mauritius) or see Newson (on cortisol levels and bonding at mass football events) or Kavanagh on Ju Jitsu belt whipping ceremonies. Newson (2016) and Buhrmester (2018) directly measure personal transformation following intense, dysphoric

experiences and how they bond participants together, including subsequent altruism in Buhrmester’s paper.

Thank you for pointing us toward some other relevant research we’d missed in our original literature review. We have now included these citations in our introductory text. Specifically, we say (p. 3):

Support for our hypothesis comes from research showing that collective gatherings such as rituals⁵², ceremonies⁵³, raves⁵⁴, and sporting events^{55,56} are associated with feelings of self-expansion and “group identity fusion”^{57,58,59,60}—a psychological state characterized by intense feelings of merging or oneness between the self and the group^{61,62}. This phenomenon is well-documented across cultures⁶³ and is associated with endorsement of group values^{64,65}, enhanced group loyalty^{66,67}, increased generosity^{68,69,70,71,72,73,74}, and cooperation^{75,76,77,78}.

- ‘Notably, concern for others beyond one’s family and community is a core feature of modern secular morality in Western democratic cultures’ – are the authors stating that this is the case specifically for Western democratic cultures and not elsewhere? Or do they refer to these cultures because that’s where the study is based? It seems to me that many pre-agricultural societies were/are concerned with life beyond one’s community, e.g., nature and a more holistic view of ‘us’ that incorporates humanity.

We apologize for the lack of clarity with this line. We do not mean to suggest that Western culture is the only one with such universalistic concerns; merely that transformative experiences that take place in cultures that emphasize such concerns may be particularly likely produce the moral expansion we observed in our data. We have removed it from the introduction text.

P4

- Is the present paper related to the following, if so, why is the latter not mentioned? For instance, the authors state that drug use is controlled throughout to focus on experiences that arose without chemical intervention (surely the reader ought to know that the two studies are connected and indeed that psychedelics also lead to TEs): <https://www.pnas.org/content/117/5/2338.short>

Due to an oversight, we mistakenly omitted this paper from our references list. We apologize for the error. We also agree that a more thorough description of the relationship between the current findings and the psychological consequences of psychedelic drug use is warranted. To that end, we have included the following in the study introduction (p. 4):

We designed a variety of experimental procedures and methods for the mass gathering context, including measures of generosity and moral expansion that did not rely on the use of money, which was prohibited at some of our field sites (see SOM 4 for full descriptions of measures and verbatim participant instructions). We also collected detailed measures of participants’ use of psychoactive substances, as past work has implicated psychedelic substance use in transformative experience and prosocial behavior^{6,79,80}. In related work, we have examined the emotional consequences of

psychedelic substance use at mass gatherings⁸¹, finding that their use is associated with increased positive mood—an effect mediated by transformative experience and universal connectedness. Here, we sought to examine the nature and prosocial correlates of transformative experiences that do not necessarily arise from psychedelic substance use. To this end, we controlled for substance use in all analyses of onsite data, and compared the psychological qualities of transformative experience that arose in the presence versus absence of psychedelic substances (see SOM 1.2 for further details).

In the Discussion, we write (p. 18):

Our research highlights both parallels and distinctions between the psychological effects of transformative experiences at secular mass gatherings and those triggered by the use of psychedelic substances. We previously reported that psychedelic substance use at mass gatherings predicts experiences of personal transformation, and that such experiences are associated with universal connectedness and positive mood⁵⁰. Yet here we find that transformative experiences at mass gatherings increase over time over and above that expected by psychedelic substance use. In supplementary analyses, we also explored the effects of psychedelics on psychological qualities of transformative experience (see SOM 1.2). While, controlling for substance use, feeling more socially connected to community, culture or history and perceiving something new about others were the most prevalent qualities of transformation reported in the onsite sample, psychedelic substance use was most strongly associated with changes to perceptions of reality and oneself, and least strongly associated with connection to community, culture, or history. This suggests that, while the use of psychedelic substances undoubtedly plays a significant role in eliciting transformative experiences⁶, such experiences might be psychologically distinct from those that emerge from mass gathering participation alone.

Please also see SOM 1.2 (reproduced in full on pp. 11-13 of this letter) for a detailed description of main effects and interactions involving drug use.

- Likewise, the use of ‘set and setting’ in the introduction seems unusual in a context that claims to control for drugs...I think it would help the reader to mention that set and setting of drugs is important in lasting outcomes, so too might set and setting of non-chemically induced experiences too.

We agree and have made the following changes to the language in the Introduction (p. 4):

Finally, we investigated whether the “set and setting” of transformative experiences is associated the nature of prosocial change^{82,83}. While these concepts were originally conceived to understand the psychological consequences of psychedelic substance use⁸⁴, we surmised they may have similarly meaningful consequences for mass gathering participation alone.

- Fig. 1

It might be helpful to have red/grey/blue/yellow/grey in the key, rather than (lost) in the legend.

We appreciate you pointing out that this figure was a bit difficult to read. We've adjusted the image to make it more intuitive. Thank you for this suggestion (p. 7)!

Results

- Is it surprising that people who have been at the event for 4+ days have had more TEs than those there for less time, they've just had more time for them? Assuming this is a between-subject analysis, people who dive right in for a science experiment have less TEs, whereas those who don't bother with a science experiment the day they arrive may presumably be having more TEs? This could therefore be character/personality driven.

Thank you for providing us an opportunity to clarify this point. We certainly agree that it's not particularly surprising that people experienced more TEs over time at these mass gatherings. However, we think it is important to report this result for several reasons. First, a basic assumption of our work is that people will be more likely to experience TEs at mass gatherings over time. Thus this result is an important validation of a key assumption of our research. Furthermore, the changes in TE over time help us to shed light on the question of whether there is something psychologically important about these events or whether the population we surveyed merely has a high propensity to say they were transformed. The fact that TEs increase over time suggests that there is something about mass gathering attendance rather than just the population sample that is causing increased reports of transformative experience over time.

We also appreciate your concern about possible selection effects being responsible for the effects we observed. On one level, we think it is unlikely that selection effects you describe could entirely explain these results, since it would need to be the case that the effect of having a transformative experience on willingness to participate in a science experiment would be different at the end of the event than at the beginning, and it seems more likely that whatever effects transformative experiences have on participation would be consistent across time.

However, we do think this is an important potential limitation to be aware of. Accordingly, we have added the following text in the Discussion (p. 18):

It is also possible that, because participants were not randomly assigned to participate, but instead volunteered, that self-selection effects could have led us to over- or under-sample participants having a transformative experience.

We also appreciate your point that personality and/or individual difference may have a significant impact on how mass gathering attendance results in prosocial change. Unfortunately, due to time constraints we did not collect such measures in our surveys; however, we certainly think such questions would be a valuable topic for future research. We have now revised our discussion to include this point. Specifically, we say (p. 18):

We also did not collect data on personality traits, like openness to experience, that might moderate these results in important ways. Investigating how personality traits interact with behavior at mass gatherings to produce transformative experiences will be an important topic for future research.

- It's also interesting that TEs go up after 6m – this fits with Whitehouse's work on the need for reflection (Whitehouse and Lanman, 2014) following an event to develop personal transformation (and subsequent group fusion), see e.g., Jong et al., 2016.

This is a very useful observation; thank you so much for bringing it to our attention! We have now added language noting this observation in the Discussion. Specifically, we say (p. 18):

we observed a positive relationship between transformative experience and generosity in the follow-up surveys conducted 1-4 weeks and 6 months following event attendance. This delayed onset of generosity following transformative experiences is consistent with research showing that highly intense social experiences tend to be followed by periods of reflection in which the personal significance of such events is consolidated into the self-concept^{10,42}. In this way, our research extends previous work on the sacrifices made by individuals on behalf of others following instances of personal transformation and subsequent feelings of identity fusion^{9,32}. Further work is necessary to investigate the longer-term trajectories of transformative experience and prosocial change.

P9:

- consider rephrasing 'common intuition' to 'One reason...', common intuition is begging for someone to disagree! For instance, people may participate not because they strongly desire/expect to be transformed, but to reconfirm their previous transformative experiences.

We have changed the text (p. 9):

It is possible that people participate in mass gatherings with strong expectations and desires to be transformed.

- Under anticipating transformation, could you state whether or not this is within-subject and longitudinal or was this at times a retrospective measure? There are a lot of measures

and it will help with the narrative to mention this briefly. If it's retrospective (which it seems to be from the SI), then this is curious – wouldn't the TE (or non TE) influence the ppt's perception of whether they had been expecting it or not? Especially if they hadn't been asked to consider their desires and anticipations and remember them in advance.

We have clarified the text accordingly (p. 10):

To test this question, we tested in a cross-sectional analysis whether expectations and desires for transformation predicted reported transformative experience.

We also agree that, ideally, information on expectations and desires would have been collected at a different time point. To make this limitation of the data clear to readers, we have now included the following text (pp. 10-11):

It should be noted that, because these measures were collected at the same time as reports on transformative experience, it is possible that transformative experiences onsite subsequently increased reports of expectations and desires for one. Some evidence that speaks against this possibility is that self-reported expectations ($M = 3.19$, $SD = 1.91$) and desires ($M = 3.71$, $SD = 2.09$) collected onsite were, if anything, lower than those collected in the pre-attendance sample ($M = 4.06$, $SD = 1.98$ and $M = 5.38$, $SD = 1.86$; $t(548) = 11.8$, $p < .001$ and $t(629) = 14.4$, $p < .001$, respectively). We cannot, however, rule out the possibility that transformative experience impacted reports of anticipation.

- 'Of the 49.6% of participants who did not even somewhat expect to be transformed, 46.7% reported being at least somewhat transformed.' This is purely descriptive – can you put some inferential statistic on this? E.g., chi squared and treat the vars as binary (grouped around the mean?) – those who did not expect to be transformed, I suspect, were significantly no less likely to be transformed than those who did expect to be transformed? - Same for desires.

You are correct in your suspicion that people who did not expect to be transformed were less likely to report being transformed. The best way of capturing this relationship, we think, is through the linear relationships reported in the previous paragraph (p. 10):

both expecting ($B = 0.15$, $SE = 0.04$, $t(1175) = 4.33$, $p < .001$) and desiring ($B = 0.23$, $SE = 0.03$, $t(1178) = 7.00$, $p < .001$) a transformative experience positively predicted participants' actually having one.

By contrast, the primary aim of the paragraph you refer to is to show that sizable percentages of people who did not expect or desire to be transformed were nevertheless at least somewhat transformed. For this, we believe descriptive statistics are sufficient. However, we agree with you that we did not make this sufficiently clear in the original text; we've now revised the text to more accurately convey our intent (p. 11):

Notwithstanding the strong relationship between the likelihood of expecting or desiring a transformative experience and the likelihood of having one, there was some

evidence the transformative experiences we observed onsite over time were not merely due to the self-fulfilling effects of anticipation. First, we found that, of the 49.6% of participants who did not even somewhat expect to be transformed, nearly half (46.7%) reported being at least somewhat transformed. Similarly, of the 41.5% of people who did not even somewhat desire to be transformed, 42.2% reported being at least somewhat transformed. These results show that considerable proportions of people who neither expected nor desired to be transformed nevertheless reported a transformative experience, thereby supporting the possibility that anticipation is not a necessary condition for transformation.

P10:

- How does drug intake interact with TEs in relation to desired/anticipated transformation? You mention that you control for drugs, presumably drug consumption is entered as a covariate (I'm reading this article linearly so I expect you'll go on to explain this in methods) – what if people who take drugs do not have their expectations met simply because their experiences go out of their control?

Thank you for encouraging us to explore drug use more thoroughly. As we wrote above (p. 12-14 of this letter), we've addressed these questions in an extended section in the SOM 1.2.

- On P4 of SM, drugs are described as being controlled for in all analyses. Drugs are a big missing step in this paper. The subject matter – TEs that are not drug induced is important and interesting in its own right but (a) why did the authors not just select a site that didn't have drugs at its forefront, e.g., Extinction Rebellion events; and (b) why not also look at the drugs (explaining that there is a sister paper would help with transparency, if this is the case). If the authors explained precisely how drugs were controlled up front, that would help. There is a line that is a little misleading, 'focus on TEs that arose without chemical intervention' – which sounds more like you excluded users of psychoactive substances.

Thank you for encouraging us to clarify how these results relate to the use of psychoactive substances. First off, as we mentioned above, an editorial mistake led us to inadvertently omit a citation from accompanying research examining the effects of psychedelics on wellbeing; we have rectified the error. We do wish to emphasize that the focus of the current research is *not* transformative experiences induced by drugs; indeed, as our analysis in SOM 1.2 shows, the nature of psychedelic-induced transformative experience seems to be distinctive from the transformative experiences reported in the full sample, controlling for all substance use. Nevertheless, we think that reporting more information on how we accounted for drug use in our dataset clarifies the relationship between our past and present findings.

We now explain in greater detail how we controlled for substance use. First, we changed the text in the line that you pointed out (p. 4):

We designed a variety of experimental procedures and methods appropriate for the mass gathering context, including measures of generosity and moral expansion that did not rely on the use of money, which was prohibited at some of our field sites (see SOM 4 for full descriptions of measures and verbatim participant instructions). We also

collected detailed measures of participants' use of psychoactive substances, as past work has implicated psychedelic substance use in transformative experience and prosocial behavior^{6,85,86}. In related work, we have examined the emotional consequences of psychedelic substance use at mass gatherings⁸⁷, finding that their use is associated with increased positive mood—an effect mediated by transformative experience and universal connectedness. Here, we sought to examine the nature and prosocial correlates of transformative experiences that do not necessarily arise from psychedelic substance use. To this end, we controlled for substance use in all analyses of onsite data, and compared the psychological qualities of transformative experience that arose in the presence versus absence of psychedelic substances (see SOM 1.2 for further details).

In our Analysis section, we include the following text (p. 21):

Participants were reminded that their responses were entirely anonymous and confidential. They were given a list of a variety of substance categories: hallucinogens, euphorics, stimulants, narcotics, and cannabis. In each category was at least one legal substance (e.g. salvia, ephedrine, kratom) rendering participants' responses non-self-incriminating. For each substance category, participants indicated whether they were currently under the influence, had taken in the last 24 hours, had taken at all that week, and had taken for the 1st time that week. Participants were binary-coded as using psychoactive substances if they selected any of those responses.

And under “analytic approach” (p. 23):

Component path analysis (that is, direct relationships between the predictor, the mediator, and the dependent variables) included field site as a random intercept and controlled for demographics (i.e., gender, age, education, religiosity, and income), and “incidental variables” (i.e., mood, expectations and desires of having a transformative experience, and the use of psychoactive substances [binary-coded as -.5 or .5 and including euphorics, hallucinogens, stimulants, alcohol, narcotics, and cannabis]).

In addition, we have now included a complete supplementary section (SOM 1.2) further exploring the relationships of drugs (see pp. 11-13).

- Drugs are surely a massive variable between the online covid event and the onsite events. Did the online covid event have music/dance or other synchronised activities that generate TEs? Did the online event have sleep deprivation? Did people attending online have relational groups attending with them? As it stands, this control group is not a meaningful comparison. Having just read the SI, it wasn't immediately clear from the main text that the covid event was also a burning man event.

We agree with you that we did not sufficiently outline the limitations of this supplementary study. As written above, we've now included the following limitations section (SOM p. 26)

1.6.4. Limitations. *The purpose of this study was to assess differences in the level of transformative experience among Burning Man attendees at a virtual versus in-person event. Our results did show such differences, suggesting that characteristics of the sample population are not solely responsible for the high rates of transformative experience reported onsite. At the same time, there are a large number of other factors that varied between the groups that limit the conclusions that can be drawn from this study, including drug use, behavioral synchrony (or any physical motion), sleep deprivation, physical hardship, physical contact with others, dancing, singing, shared food and rituals around eating and food preparation, etc. Thus it is impossible to specify what, exactly may have led to the greater prevalence of transformative experience at the onsite gathering. Further research will be needed to better understand the psychological differences between virtual and in-person events.*

We have also clarified in the main text that the study took place among Burning Man attendees (p. 5).

we collected comparison data (n = 98) from a “virtual” mass gathering that took place among Burning Man attendees in the context of the Covid-19 pandemic (see SOM 1.6 for details).

- ‘This suggests that the majority of self-reported aspects of mass gathering attendance are consistent regardless of local event norms.’ – I disagree, the only local event norm measured was presence of a gift economy so this should be rephrased more specifically. The other festivals may also have more of a gift /trade economy than conventional/mainstream society albeit with the presence of cash. Local event norms at these festivals could also include, for example, freedom of expression, openness to drug taking, creativity etc. So mass gatherings that varied in these facets might produce differing degrees of TEs, but overall the gatherings studied were relatively similar.

Thank you for this clarification! We have changed the text accordingly (p. 11):

This suggests that the majority of self-reported aspects of mass gathering attendance are consistent regardless of whether the event operates a gift economy.

Fusion:

- A major problem here is that the targets for the IOS and fusion scales differed so the two cannot be compared (and these targets are not explicitly noted in the body of the text from my reading). Fusion is defined as connection to attendees compared to a larger group. Fusion is not just relationally different but different in its depth, it is a more visceral and enduring bonding – so the final option of fusion (total immersion) is categorically different from connectedness measured by the IOS. This should be stated for readers not familiar with the fusion literature.

Thank you for this clarification. We chose to use the response format of the original IOS and identity fusion scales so that each set of results could be comparable with the past extensive literature using these scales, but we also recognize that the difference you highlighted limits the

extent to which the measures can be directly compared with one another. We now note this in the text (p. 12):

It is important to note that universal connectedness and the group identity fusion scales differed in their extremity, with the latter indicating “total immersion” of the self in the group, while the former indicated only a high degree of overlap, so these measures cannot be compared directly.

We have also added a description of the “group identity fusion” measure in the “Materials” section (p. 22):

Group identity fusion. *We employed a measure developed in previous research⁹ that shows participants a small circle and a large circle representing the self and the group, respectively. Circles are presented in five consecutive iterations ranging from completely nonoverlapping to the small circle being completely subsumed by the large circle and asked to indicate which set of circles best captured their relationship to the group.*

We also include the additional information in the description of universal connectedness (p. 22):

participants are presented with series of progressively overlapping sets of circles...ranging from nonoverlapping to almost entirely overlapping, and asked to indicate which best describes their relationship...to “other human beings, in general.”

- Perhaps it’s bonding to humanity that drove moral expansion more than bonding to friends, rather than connectedness over fusion. Certainly, this is what would be suggested by fusion theory (see, for instance, Swann et al., 2009; 2012 or Whitehouse 2013 – three wishes for the world).

We certainly agree that this is a possibility! Indeed, we suspect that our use of the word “social connectedness” conveyed the erroneous impression that we see the results as contradicting the notion that “bonding with all humanity” is driving the results. In response to this point, in order to clarify our perspective on this phenomenon and its relationship to existing literature, we have replaced the term “social connectedness” with “universal connectedness” throughout the paper. Our goal here is to make clear the primary finding: namely, that feelings of connection or fusion with all human beings, as opposed to merely with members of one’s group, is driving our effects. In the results, we also now cite the papers you mentioned to clarify that our results may be considered congruent with other research on a more global form of identity fusion (p. 13):

these results are consistent with the idea that prosocial change at multi-day mass gatherings may be the result of a form of universal identity fusion characterized by connectedness to all humanity^{37,88,89}.

- What happens when social connectedness is subbed out for fusion in the P12 analyses?

As we now report in the main text results (p. 13) and the SOM 1.12.1, we find that, when substituting group identity fusion for social connectedness (now named universal connectedness) group identity fusion does not significantly predict moral expansion or mediate the relationship between time and moral expansion (p. 13):

we explored the possibility that the relationship between time and moral expansion was mediated by “group identity fusion”—a variable reflecting people’s sense of overlap with other event attendees as opposed to other humans in general. First, we substituted group identity fusion for universal connectedness in the model predicting moral expansion from time onsite and transformative experience (see Figure 4B). Group identity fusion did not significantly predict moral expansion ($B = 0.06$, $SE = 0.04$, $t(809) = 1.50$, $p = .135$ versus $B = 0.09$, $SE = 0.03$, $t(960) = 3.43$, $p = .001$ for universal connectedness), and the indirect effect of time onsite to moral expansion through group identity fusion was not significant, $B = 0.003$, $SE = 0.002$, $p = 0.155$, $CI_{95}[-0.001, 0.006]$. Moreover, a model that allowed both variables to compete for variance by including both as covariates showed that universal connectedness predicted moral expansion even when controlling for group identity fusion ($B = 0.09$, $SE = 0.03$, $t(798) = 3.13$, $p = .002$). These results suggests that changes in moral expansion are due more to generalized increases in universal connectedness than to increased connection other eventgoers(see SOM 1.12).

P12:

- precisely how many had received a ‘full dose’ of a gathering, i.e., what % attended for more than a day (there are always dropouts).

Results showed that 14 of the 1759 participants who responded to this measure (0.7%) stayed for 1 day or less. We have noted this in the results section (p. 13):

descriptive analysis shows that 99.3% of the sample indicated they stayed for more than 1 day.

- What might explain the relationship between TEs and generosity over time. I’d urge the authors to reconsider the role of identity fusion. Work by Whitehouse (2017; 2018) and colleagues suggests that fusion takes time to emerge, via a process of reflection (Jong et al., 2015) and personal transformation (Newson et al., 2016)...this might explain why prosocial behaviour’s relationship with transformation did not immediately appear, rather it appeared once the group was internalised as part of one’s core identity via identity fusion.

As we remarked above, we are very grateful to you for bringing this highly relevant work to our attention! We’ve now noted this in the study Discussion (p. 17):

This delayed onset of generosity following transformative experiences is consistent with research showing that highly intense social experiences tend to be followed by periods of reflection in which the personal significance of such events is consolidated into the self-concept^{10,42}. In this way, our research extends previous work

on the sacrifices made by individuals on behalf of others following instances of personal transformation and subsequent feelings of identity fusion^{9,32}. Further work is necessary to investigate the longer-term trajectories of transformative experience and prosocial change.

- The authors seem to point toward global implications and the value of TEs, which brings the paper back to the start. What actually causes a TE???

We agree that the question of what causes transformative experiences is an important one, worthy of its own investigation. However, this was not the main focus of the present study. Instead, our goal was to investigate the psychological qualities of transformative experiences as they are occurring, and to examine the prosocial correlates of those experiences. To address our central question, we collected data at field sites where we expected a high prevalence of transformative experiences (since these are so difficult to elicit in lab settings), and examined how transformative experiences are predictive of prosocial attitudes behavior.

At the same time, we do have some data that speaks to the fascinating question of what causes transformative experiences at mass gatherings. We conducted exploratory analyses that are now referenced in the results section of the main text:

***Predictors of transformative experience.** Next, we examined demographic, affective, and behavioral predictors of transformative experience. In our main models, we did not observe associations between transformative experience and gender, age, or income (all p s > .3). Transformative experience was negatively associated with educational attainment, $B = -0.12$, $SE = 0.05$, $t(1177) = -2.52$, $p = .012$ and the consumption of alcohol, $B = 0.29$, $SE = 0.13$, $t(1177) = -2.24$, $p < .025$. Transformative experience was positively associated with mood, $B = 0.33$, $SE = 0.06$, $t(1178) = 5.19$, $p < .001$, and the use of psychedelic substances, $B = 0.37$, $SE = 0.13$, $t(1177) = 2.75$, $p = .006$. We also built exploratory models to examine the contribution of additional behavioral variables to transformative experiences. These analyses suggested that increased reports of transformative experiences over time could be partially attributed to the formation of new social relationships, gift exchange, and dancing (see SOM 1.13 for details).*

The full exploratory analyses can be found in a new section in the SOM (1.13) that describes in full additional behavioral variables that significantly predict transformative experience, and that provide promising opportunities for future investigations into the causes of transformative experiences at mass gatherings (p. 33):

1.13. Predictors of Transformative Experience

Here we conduct exploratory analyses of several variables that might help to explain how mass gathering participation engenders transformative experiences. In the following analyses, we specified models that took a variety of behaviors and tested via mediation whether they helped account for the increase in transformative experience over time.

1.13.1. New friends. First, we tested whether the formation of new social relationships was associated with transformative experience over time. Participants indicated how many new friends they had made thus far at the event (0 to More than 50). A mediation model with time as the predictor, new friends as the mediator, and transformative experience as the dependent variable showed a significant indirect effect, $B = 0.06$, $SE = 0.009$, $p < 0.001$, $CI_{95}[0.042, 0.078]$, suggesting that forming new social connections partially accounted for the increase in the prevalence of transformative experience over time. However, the direct effect of time onsite on transformative experience remained significant in the model, $B = 0.1$, $SE = 0.025$, $p < 0.001$, $CI_{95}[0.051, 0.15]$, suggesting that this behavior did not entirely explain this relationship.

1.13.2. Giving and receiving gifts. Next we tested whether participation in gift exchange helped to account for the increase in transformative experience over time. Participants were asked how many gifts they had given (0 to More than 50) and received (0 to More than 50) thus far at the event. We then examined the indirect effect of time on transformative experience through each of these variables respectively. Results showed significant indirect effects of both giving, $B = 0.071$, $SE = 0.012$, $p < 0.001$, $CI_{95}[0.048, 0.094]$, and receiving, $B = 0.068$, $SE = 0.011$, $p < 0.001$, $CI_{95}[0.046, 0.09]$, gifts. However, in both cases ($B = 0.089$, $SE = 0.027$, $p = 0.001$, $CI_{95}[0.037, 0.142]$; $B = 0.092$, $SE = 0.027$, $p = 0.001$, $CI_{95}[0.04, 0.145]$) the direct effects from time to transformative experience remained significant when including these behaviors in the model, suggesting that these behaviors did not entirely account for the relationship between time and transformative experience.

1.13.3. Dancing (behavioral synchrony). Finally, following research suggesting that behavioral synchrony can engender generalized prosocial behavior⁹⁰, we examined whether self-reported frequency of dancing accounted for the relationship between time and transformative experience. Participants indicated how many separate occasions they had danced at the event name so far (0 to More than 20). As with the other variables, analysis showed a significant indirect effect, $B = 0.041$, $SE = 0.009$, $p < 0.001$, $CI_{95}[0.023, 0.059]$, yet the direct effect of time on transformative experience remained significant when including it in the model, $B = 0.119$, $SE = 0.026$, $p < 0.001$, $CI_{95}[0.068, 0.171]$, suggesting it did not entirely account for the effect.

We also now highlight in the Discussion the importance of understanding what causes transformative experiences, existing research that speaks to this question, and the need for additional research on the topic (p. 19):

it remains unclear exactly which aspects of mass gathering attendance cause transformative experiences. While our study was not primarily designed to answer this question, our analyses suggest that activities such as dancing (which can elicit emotional synchrony^{17,21,29}), giving and receiving gifts (which may amplify prosocial emotions⁹¹), and making new friends (which may increase feelings of mutual obligation^{18,7,9}) partially mediated the relationship between time onsite and transformative experience (SOM 1.13 for more details). Other research has identified sleep deprivation and exposure to powerful rhythmic music as additional causes of transformative experience⁶. Additional research will be needed to better understand how these factors interact with aspects of the local event culture and individual differences to produce prosocial transformation.

Methods

- There are a few typos in methods, e.g., ‘measures’ / ‘in category was at least one’

Thank you for bringing these to our attention; they have been fixed.

References

The Whitehouse & Lanman (2014) reference in Google Scholar is incorrect, as far as I know and only these two authors were on the original paper.

Thank you, we have revised the citation.

Reporting summary

- **It is very surprising that ppts scored so low for liberalism at what I would perceive to be pretty liberal festivals!**

Here is what we wrote in the Reporting Summary:

The sample skewed liberal, with a mean of 2.6 (“somewhat liberal”) on a 7-point scale (1 = “extremely liberal”, 7 = “extremely conservative”).

The mean value here seems quite liberal to us. Are we misunderstanding your point?

- **What exactly were the ‘prizes’ offered to ppts?**

Thanks for asking for clarification. As we now write in the Methods (p. 22):

the mystery box contained a variety of items that would be of value to eventgoers, including ring pops, snap bracelets, and earplugs.

SI

- **The authors report several additional online studies in the SI. Presuming these studies are not published elsewhere, these are studies worthy of more credit and could perhaps be mentioned in the abstract (‘a series of on site and X online studies’).**

Thank you for encouraging us to highlight this additional work! We’ve added the following line to the abstract:

In 6 field studies and 22 online followup studies spanning 5 years...

- **However, the SI is extensive. Take SI 1.4 – this reports a whole study with a discussion. Its connection to the main text is not entirely clear, it seems like it needs a space of its own. Was this study integral to the design of the main study, or a failed study that was still worthy of being written up?**

This study built on some of our longitudinal data to provide a clearer picture of how transformative experience related to prosocial change over time. The analyses suggest that, for example, the greater participants' reported level of transformative experience after attending these mass gatherings, the greater their change in universal connectedness before versus after the event. At the same time, we have determined that these data, due both to their limited sample size and mixed results, do not provide sufficient clarity on the nature of our effects to warrant inclusion in the main paper.

We wanted to make these results available to readers so that those who want to can obtain a more detailed understanding of the phenomena in question. However, we agree that we didn't make the connection to the studies explained in the main paper adequately clear. We now include the following section in the main results section to highlight the connection between this supplementary section and the main results (p. 14):

In supplementary analyses, we tested a hypothesis that transformative experience reported onsite would predict moral expansion and universal connectedness following attendance in the same participants, 1-4 months and 6 months later. Although we found some evidence supporting the hypothesis, ultimately the longitudinal sample was not sufficiently powered to permit firm conclusions regarding the long-term effect of transformative experience on prosocial change (SOM 1.4).

- Could the authors condense the text into tables perhaps? E.g. giving %s to Likert type scales in written form is lengthy and a table might be quicker for the reader to extract info from.

Thank you for the suggestion! We have converted the information conveying drug use into a table (SOM p. 5).

- P28 seems to have referencing errors.

We have rectified the error, thank you.

REFERENCES

- ¹. Biela, A., & Tobacyk, J. J. (1987). Self-transcendence in the agoral gathering: A case study of Pope John Paul II's 1979 visit to Poland. *Journal of Humanistic Psychology*, 27(4), 390-405.
- ². Kozinets, R. V., & Sherry Jr, J. F. (2004). Dancing on common ground: exploring the sacred at burning man. *Rave culture and religion*, 287-303.
- ³ Paul, L. A. (2014). *Transformative experience*. OUP Oxford.
- ⁴. Fischer, R., Xygalatas, D., Mitkidis, P., Reddish, P., Tok, P., Konvalinka, I., & Bulbulia, J. (2014). The fire-walker's high: Affect and physiological responses in an extreme collective ritual. *PLoS one*, 9(2), e88355.
- ⁵. Hopkins, N., Reicher, S. D., Khan, S. S., Tewari, S., Srinivasan, N., & Stevenson, C. (2016). Explaining effervescence: Investigating the relationship between shared social identity and positive experience in crowds. *Cognition and Emotion*, 30(1), 20-32.
- ⁶. Newson, M., Khurana, R., Cazorla, F., & Mulukom, V. (2021). I get high with a little help from my friends' - How raves can invoke identity fusion and lasting co-operation via transformative experiences. *Frontiers in Psychology*.
- ⁷. Newson, M., Buhrmester, M., & Whitehouse, H. (2016). Explaining lifelong loyalty: The role of identity fusion and self-shaping group events. *PLoS one*, 11(8), e0160427.
- ⁸. Newson, M. (2019). Football, fan violence, and identity fusion. *International Review for the Sociology of Sport*, 54(4), 431-444.
- ⁹. Swann Jr, W. B., Jetten, J., Gómez, Á., Whitehouse, H., & Bastian, B. (2012). When group membership gets personal: A theory of identity fusion. *Psychological review*, 119(3), 441.
- ¹⁰. Whitehouse, H., & Lanman, J. A. (2014). The ties that bind us: Ritual, fusion, and identification. *Current Anthropology*, 55(6), 674-695.
- ¹¹. Besta, T., Jaśkiewicz, M., Kosakowska-Berezecka, N., Lawendowski, R., & Zawadzka, A. M. (2018). What do I gain from joining crowds? Does self-expansion help to explain the relationship between identity fusion, group efficacy and collective action?. *European Journal of Social Psychology*, 48(2), 0152-0167.
- ¹². Reese, E., & Whitehouse, H. (2021). The development of identity fusion. *Perspectives on Psychological Science*, 1745691620968761.
- ¹³ Kavanagh, C. M., Kapitány, R., Putra, I. E., & Whitehouse, H. (2020). Exploring the pathways between transformative group experiences and identity fusion. *Frontiers in Psychology*, 11, 1172.
- ¹⁴ Aron, A., Aron, E. N., & Smollan, D. (1992). Inclusion of other in the self scale and the structure of interpersonal closeness. *Journal of personality and social psychology*, 63(4), 596.
- ¹⁵. Fiske, A. P. (2019). *Kama muta: Discovering the connecting emotion*. Routledge.
- ¹⁶. Drury, J., & Reicher, S. (2005). Explaining enduring empowerment: A comparative study of collective action and psychological outcomes. *European Journal of Social Psychology*, 35(1), 35-58.
- ¹⁷. Páez, D., Rimé, B., Basabe, N., Włodarczyk, A., & Zumeta, L. (2015). Psychosocial effects of perceived emotional synchrony in collective gatherings. *Journal of Personality and Social Psychology*, 108(5), 711.
- ¹⁸. Hopkins, N., Reicher, S. D., Khan, S. S., Tewari, S., Srinivasan, N., & Stevenson, C. (2016). Explaining effervescence: Investigating the relationship between shared social identity and positive experience in crowds. *Cognition and Emotion*, 30(1), 20-32.
- ¹⁹. Khan, S. S., Hopkins, N., Reicher, S., Tewari, S., Srinivasan, N., & Stevenson, C. (2016). How collective participation impacts social identity: A longitudinal study from India. *Political Psychology*, 37(3), 309-325.
- ²⁰ Batson, C. D., Sager, K., Garst, E., Kang, M., Rubchinsky, K., & Dawson, K. (1997). Is empathy-induced helping due to self-other merging?. *Journal of personality and social psychology*, 73(3), 495.
- ²¹ Reddish, P., Tong, E. M., Jong, J., Lanman, J. A., & Whitehouse, H. (2016). Collective synchrony increases prosociality towards non-performers and outgroup members. *British Journal of Social Psychology*, 55(4), 722-738.
- ²². Buhrmester, M. D., Fraser, W. T., Lanman, J. A., Whitehouse, H., & Swann, W. B. (2015). When terror hits home: Identity fused Americans who saw Boston bombing victims as "family" provided aid. *Self and Identity*, 14, 253-270.
- ²³. Xygalatas, D., Mitkidis, P., Fischer, R., Reddish, P., Skewes, J., Geertz, A. W., Roepstorff, A., & Bulbulia, J. (2013). Extreme rituals promote prosociality. *Psychological science*, 24(8), 1602-16

-
- ²⁴. Buhrmester, M. D., Burnham, D., Johnson, D. D., Curry, O. S., Macdonald, D. W., & Whitehouse, H. (2018). How moments become movements: Shared outrage, group cohesion, and the lion that went viral. *Frontiers in Ecology and Evolution*, 6, 54.
- ²⁵. Misch, A., Fergusson, G., & Dunham, Y. (2018). Temporal dynamics of partisan identity fusion and prosociality during the 2016 US presidential election. *Self and Identity*, 17(5), 531-548.
- ²⁶. Segal, K., Jong, J., & Halberstadt, J. (2018). The fusing power of natural disasters: An experimental study. *Self and Identity*, 17, 574-586
- ²⁷. Sosis, R., & Ruffle, B. J. (2003). Religious ritual and cooperation: Testing for a relationship on Israeli religious and secular Kibbutzim. *Current Anthropology*, 44(5), 713-722
- ²⁸. Fischer, R., Callander, R., Reddish, P., & Bulbulia, J. (2013). How do rituals affect cooperation?. *Human Nature*, 24(2), 115-125.
- ²⁹. Jackson, J. C., Jong, J., Bilkey, D., Whitehouse, H., Zollmann, S., McNaughton, C., & Halberstadt, J. (2018). Synchrony and physiological arousal increase cohesion and cooperation in large naturalistic groups. *Scientific reports*, 8(1), 127.
- ³⁰. Whitehouse, H., Jong, J., Buhrmester, M. D., Gómez, Á., Bastian, B., Kavanagh, C. M., ... & Gavrillets, S. (2017). The evolution of extreme cooperation via shared dysphoric experiences. *Scientific Reports*, 7(1), 1-10.
- ³¹. Swann Jr, W. B., Buhrmester, M. D., Gómez, A., Jetten, J., Bastian, B., Vázquez, A., ... & Zhang, A. (2014). What makes a group worth dying for? Identity fusion fosters perception of familial ties, promoting self-sacrifice. *Journal of personality and social psychology*, 106(6), 912.
- ³² Whitehouse, H. (2018). Dying for the group: Towards a general theory of extreme self-sacrifice. *Behavioral and Brain Sciences*, 41.
- ³³. Singer, P. (1981). *The expanding circle*. Oxford: Clarendon Press
- ³⁴. Crimston, D., Hornsey, M. J., Bain, P. G., & Bastian, B. (2018). Toward a psychology of moral expansiveness. *Current Directions In Psychological Science*, 27(1), 14-19.
- ³⁵. McFarland, S., Webb, M., & Brown, D. (2012). All humanity is my ingroup: A measure and studies of identification with all humanity. *Journal of Personality and Social psychology*, 103(5), 830.
- ³⁶ Pizarro, J. J., Basabe, N., Fernández, I., Carrera, P., Apodaca, P., Ging, C. I. M., ... & Páez, D. (2021). Self-Transcendent Emotions and Their Social Effects: Awe, Elevation and Kama Muta Promote a Human Identification and Motivations to Help Others. *Frontiers in psychology*, 12.
- ³⁷ Stellar, J. E., Gordon, A. M., Piff, P. K., Cordaro, D., Anderson, C. L., Bai, Y., Maruskin, L., & Keltner, D. (2017). Self-transcendent emotions and their social functions: Compassion, gratitude, and awe bind us to others through prosociality. *Emotion Review*, 9(3), 200-207.
- ³⁸ Yaden, D. B., Haidt, J., Hood Jr, R. W., Vago, D. R., & Newberg, A. B. (2017). The varieties of self-transcendent experience. *Review of general psychology*, 21(2), 143-160.
- ³⁹. Kapitány, R., Kavanagh, C., Buhrmester, M. D., Newson, M., & Whitehouse, H. (2019). Ritual, identity fusion, and the inauguration of president Trump: a pseudo-experiment of ritual modes theory. *Self and Identity*, 1-31.
- ⁴⁰. Khan, S. R., & Stagnaro, M. N. (2016). The influence of multiple group identities on moral foundations. *Ethics & Behavior*, 26(3), 194-214.
- ⁴¹. Doğruyol, B., Alper, S., & Yilmaz, O. (2019). The five-factor model of the moral foundations theory is stable across WEIRD and non-WEIRD cultures. *Personality and Individual Differences*, 151, 109547.
- ⁴². Buchtel, E. E., Guan, Y., Peng, Q., Su, Y., Sang, B., Chen, S. X., & Bond, M. H. (2015). Immorality East and West: Are Immoral Behaviors Especially Harmful, or Especially Uncivilized?
- ⁴³. Kim, H., & Markus, H. R. (1999). Deviance or uniqueness, harmony or conformity? A cultural analysis. *Journal of personality and social psychology*, 77(4), 785
- ⁴⁴ Paul, L. A. (2014). *Transformative experience*. OUP Oxford.
- ⁴⁵ Pokorny, T., Preller, K. H., Kometer, M., Dziobek, I., & Vollenweider, F. X. (2017). Effect of psilocybin on empathy and moral decision-making. *International Journal of Neuropsychopharmacology*, 20(9), 747-757.
- ⁴⁶ Griffiths, R. R., Johnson, M. W., Richards, W. A., Richards, B. D., Jesse, R., MacLean, K. A., Barrett, F. A., Cosimano, M. P., & Klinedinst, M. A. (2018). Psilocybin-occasioned mystical-type experience in combination with meditation and other spiritual practices produces enduring positive changes in psychological functioning and in trait measures of prosocial attitudes and behaviors. *Journal of Psychopharmacology*, 32(1), 49-69.

-
- ⁴⁷. Forstmann, M., Yudkin, D. A., Prosser, A. M., Heller, S. M., & Crockett, M. J. (2020). Transformative experience and universal connectedness mediate the mood-enhancing effects of psychedelic use in naturalistic settings. *Proceedings of the National Academy of Sciences*, *117*(5), 2338-2346.
- ⁴⁸ Pokorny, T., Preller, K. H., Kometer, M., Dziobek, I., & Vollenweider, F. X. (2017). Effect of psilocybin on empathy and moral decision-making. *International Journal of Neuropsychopharmacology*, *20*(9), 747-757.
- ⁴⁹ Griffiths, R. R., Johnson, M. W., Richards, W. A., Richards, B. D., Jesse, R., MacLean, K. A., Barrett, F. A., Cosimano, M. P., & Klinedinst, M. A. (2018). Psilocybin-occasioned mystical-type experience in combination with meditation and other spiritual practices produces enduring positive changes in psychological functioning and in trait measures of prosocial attitudes and behaviors. *Journal of Psychopharmacology*, *32*(1), 49-69.
- ⁵⁰. Forstmann, M., Yudkin, D. A., Prosser, A. M., Heller, S. M., & Crockett, M. J. (2020). Transformative experience and universal connectedness mediate the mood-enhancing effects of psychedelic use in naturalistic settings. *Proceedings of the National Academy of Sciences*, *117*(5), 2338-2346.
- ⁵¹ Forstmann, M., Yudkin, D. A., Prosser, A. M., Heller, S. M., & Crockett, M. J. (2020). Transformative experience and universal connectedness mediate the mood-enhancing effects of psychedelic use in naturalistic settings. *Proceedings of the National Academy of Sciences*, *117*(5), 2338-2346.
- ⁵². Fischer, R., Xygalatas, D., Mitkidis, P., Reddish, P., Tok, P., Konvalinka, I., & Bulbulia, J. (2014). The fire-walker's high: Affect and physiological responses in an extreme collective ritual. *PLoS one*, *9*(2), e88355.
- ⁵³. Hopkins, N., Reicher, S. D., Khan, S. S., Tewari, S., Srinivasan, N., & Stevenson, C. (2016). Explaining effervescence: Investigating the relationship between shared social identity and positive experience in crowds. *Cognition and Emotion*, *30*(1), 20-32.
- ⁵⁴. Newson, M., Khurana, R., Cazorla, F., & Mulukom, V. (2021). I get high with a little help from my friends' - How raves can invoke identity fusion and lasting co-operation via transformative experiences. *Frontiers in Psychology*.
- ⁵⁵. Newson, M., Buhrmester, M., & Whitehouse, H. (2016). Explaining lifelong loyalty: The role of identity fusion and self-shaping group events. *PLoS one*, *11*(8), e0160427.
- ⁵⁶. Newson, M. (2019). Football, fan violence, and identity fusion. *International Review for the Sociology of Sport*, *54*(4), 431-444.
- ⁵⁷. Swann Jr, W. B., Jetten, J., Gómez, Á., Whitehouse, H., & Bastian, B. (2012). When group membership gets personal: A theory of identity fusion. *Psychological review*, *119*(3), 441.
- ⁵⁸. Whitehouse, H., & Lanman, J. A. (2014). The ties that bind us: Ritual, fusion, and identification. *Current Anthropology*, *55*(6), 674-695.
- ⁵⁹. Besta, T., Jaśkiewicz, M., Kosakowska-Berezecka, N., Lawendowski, R., & Zawadzka, A. M. (2018). What do I gain from joining crowds? Does self-expansion help to explain the relationship between identity fusion, group efficacy and collective action?. *European Journal of Social Psychology*, *48*(2), O152-O167.
- ⁶⁰. Reese, E., & Whitehouse, H. (2021). The development of identity fusion. *Perspectives on Psychological Science*, *17*45691620968761.
- ⁶¹ Kavanagh, C. M., Kapitány, R., Putra, I. E., & Whitehouse, H. (2020). Exploring the pathways between transformative group experiences and identity fusion. *Frontiers in Psychology*, *11*, 1172.
- ⁶² Aron, A., Aron, E. N., & Smollan, D. (1992). Inclusion of other in the self scale and the structure of interpersonal closeness. *Journal of personality and social psychology*, *63*(4), 596.
- ⁶³. Fiske, A. P. (2019). *Kama muta: Discovering the connecting emotion*. Routledge.
- ⁶⁴. Drury, J., & Reicher, S. (2005). Explaining enduring empowerment: A comparative study of collective action and psychological outcomes. *European Journal of Social Psychology*, *35*(1), 35-58.
- ⁶⁵. Páez, D., Rimé, B., Basabe, N., Wlodarczyk, A., & Zumeta, L. (2015). Psychosocial effects of perceived emotional synchrony in collective gatherings. *Journal of Personality and Social Psychology*, *108*(5), 711.
- ⁶⁶. Hopkins, N., Reicher, S. D., Khan, S. S., Tewari, S., Srinivasan, N., & Stevenson, C. (2016). Explaining effervescence: Investigating the relationship between shared social identity and positive experience in crowds. *Cognition and Emotion*, *30*(1), 20-32.
- ⁶⁷. Khan, S. S., Hopkins, N., Reicher, S., Tewari, S., Srinivasan, N., & Stevenson, C. (2016). How collective participation impacts social identity: A longitudinal study from India. *Political Psychology*, *37*(3), 309-325.
- ⁶⁸ Batson, C. D., Sager, K., Garst, E., Kang, M., Rubchinsky, K., & Dawson, K. (1997). Is empathy-induced helping due to self-other merging?. *Journal of personality and social psychology*, *73*(3), 495.
- ⁶⁹ Reddish, P., Tong, E. M., Jong, J., Lanman, J. A., & Whitehouse, H. (2016). Collective synchrony increases prosociality towards non-performers and outgroup members. *British Journal of Social Psychology*, *55*(4), 722-738.

-
- ⁷⁰. Buhrmester, M. D., Fraser, W. T., Lanman, J. A., Whitehouse, H., & Swann, W. B. (2015). When terror hits home: Identity fused Americans who saw Boston bombing victims as “family” provided aid. *Self and Identity*, 14, 253–270.
- ⁷¹. Xygalatas, D., Mitkidis, P., Fischer, R., Reddish, P., Skewes, J., Geertz, A. W., Roepstorff, A., & Bulbulia, J. (2013). Extreme rituals promote prosociality. *Psychological science*, 24(8), 1602-16
- ⁷². Buhrmester, M. D., Burnham, D., Johnson, D. D., Curry, O. S., Macdonald, D. W., & Whitehouse, H. (2018). How moments become movements: Shared outrage, group cohesion, and the lion that went viral. *Frontiers in Ecology and Evolution*, 6, 54.
- ⁷³. Misch, A., Fergusson, G., & Dunham, Y. (2018). Temporal dynamics of partisan identity fusion and prosociality during the 2016 US presidential election. *Self and Identity*, 17(5), 531-548.
- ⁷⁴. Segal, K., Jong, J., & Halberstadt, J. (2018). The fusing power of natural disasters: An experimental study. *Self and Identity*, 17, 574–586
- ⁷⁵. Sosis, R., & Ruffle, B. J. (2003). Religious ritual and cooperation: Testing for a relationship on Israeli religious and secular Kibbutzim. *Current Anthropology*, 44(5), 713-722
- ⁷⁶. Fischer, R., Callander, R., Reddish, P., & Bulbulia, J. (2013). How do rituals affect cooperation?. *Human Nature*, 24(2), 115-125.
- ⁷⁷. Jackson, J. C., Jong, J., Bilkey, D., Whitehouse, H., Zollmann, S., McNaughton, C., & Halberstadt, J. (2018). Synchrony and physiological arousal increase cohesion and cooperation in large naturalistic groups. *Scientific reports*, 8(1), 127.
- ⁷⁸. Whitehouse, H., Jong, J., Buhrmester, M. D., Gómez, Á., Bastian, B., Kavanagh, C. M., ... & Gavrillets, S. (2017). The evolution of extreme cooperation via shared dysphoric experiences. *Scientific Reports*, 7(1), 1-10.
- ⁷⁹ Pokorny, T., Preller, K. H., Kometer, M., Dziobek, I., & Vollenweider, F. X. (2017). Effect of psilocybin on empathy and moral decision-making. *International Journal of Neuropsychopharmacology*, 20(9), 747-757.
- ⁸⁰ Griffiths, R. R., Johnson, M. W., Richards, W. A., Richards, B. D., Jesse, R., MacLean, K. A., Barrett, F. A., Cosimano, M. P., & Klinedinst, M. A. (2018). Psilocybin-occasioned mystical-type experience in combination with meditation and other spiritual practices produces enduring positive changes in psychological functioning and in trait measures of prosocial attitudes and behaviors. *Journal of Psychopharmacology*, 32(1), 49-69.
- ⁸¹. Forstmann, M., Yudkin, D. A., Prosser, A. M., Heller, S. M., & Crockett, M. J. (2020). Transformative experience and universal connectedness mediate the mood-enhancing effects of psychedelic use in naturalistic settings. *Proceedings of the National Academy of Sciences*, 117(5), 2338-2346.
- ⁸². Rosegrant, J. (1976). The impact of set and setting on religious experience in nature. *Journal for the Scientific Study of Religion*, 301-310.
- ⁸³. Hood Jr, R. W. (1978). Anticipatory set and setting: Stress incongruities as elicitors of mystical experience in solitary nature situations. *Journal for the Scientific Study of Religion*, 279-287.
- ⁸⁴. Hartogsohn, I. (2016). Set and setting, psychedelics and the placebo response: an extra-pharmacological perspective on psychopharmacology. *Journal of Psychopharmacology*, 30(12), 1259-1267.
- ⁸⁵ Pokorny, T., Preller, K. H., Kometer, M., Dziobek, I., & Vollenweider, F. X. (2017). Effect of psilocybin on empathy and moral decision-making. *International Journal of Neuropsychopharmacology*, 20(9), 747-757.
- ⁸⁶ Griffiths, R. R., Johnson, M. W., Richards, W. A., Richards, B. D., Jesse, R., MacLean, K. A., Barrett, F. A., Cosimano, M. P., & Klinedinst, M. A. (2018). Psilocybin-occasioned mystical-type experience in combination with meditation and other spiritual practices produces enduring positive changes in psychological functioning and in trait measures of prosocial attitudes and behaviors. *Journal of Psychopharmacology*, 32(1), 49-69.
- ⁸⁷. Forstmann, M., Yudkin, D. A., Prosser, A. M., Heller, S. M., & Crockett, M. J. (2020). Transformative experience and universal connectedness mediate the mood-enhancing effects of psychedelic use in naturalistic settings. *Proceedings of the National Academy of Sciences*, 117(5), 2338-2346.
- ⁸⁸ Whitehouse, H., Swann, W., Ingram, G., Prochownik, K., Lanman, J., Waring, T. M., ... & Johnson, D. (2013). Three wishes for the world. *Cliodynamics: The Journal of Theoretical and Mathematical History*, 4(2).
- ⁸⁹. Swann Jr, W. B., Gómez, A., Seyle, D. C., Morales, J., & Huici, C. (2009). Identity fusion: the interplay of personal and social identities in extreme group behavior. *Journal of personality and social psychology*, 96(5), 995.
- ⁹⁰ Reddish, P., Tong, E. M., Jong, J., Lanman, J. A., & Whitehouse, H. (2016). Collective synchrony increases prosociality towards non-performers and outgroup members. *British Journal of Social Psychology*, 55(4), 722-738.
- ⁹¹ Aknin, L. B., Dunn, E. W., Sandstrom, G. M., & Norton, M. I. (2013). Does social connection turn good deeds into good feelings?: On the value of putting the ‘social’ in prosocial spending. *International Journal of Happiness and Development*, 1(2), 155-171.

REVIEWER COMMENTS

Reviewer #1 (Remarks to the Author):

The authors did a great job in addressing all of my questions and concerns. I continue to have a very positive assessment of this work, and I think it will have a major impact on the field. I think it is ready for publication.

Reviewer #2 (Remarks to the Author):

This was a thorough revision which I felt addressed my previous concerns! I only have one further query.

The authors say (In 406) It is important to note that universal connectedness and the group identity fusion scales differed in their extremity, with the latter indicating —total immersion of the self in the group, while the former indicated only a high degree of overlap, so these measures cannot be compared directly.

This suggests they used the fusion measure as a dichotomous measures (fused=completely overlapping vs. all other partial overlap ratings)? However, this was not clear in the methods, and rather it sounded like they were treating it as a continuous measure? Theoretically it should be treated as dichotomous, and perhaps that is how it was treated, but I could not see where that was specified?

Reviewer #3 (Remarks to the Author):

This is a very well organised response, thank you!

You've addressed all of my queries well. I would recommend Newson et al., 2021 (Frontiers - for transparency, this is my paper!), which investigates the role of psychedelics on transformative experiences (which in turn appeared to be associated with identity fusion). This might be useful with the Frostman reference showing that work has already been done on this area, hence your paper not focussing on drugs. It may also be relevant to your new lines on personality traits (trait openness was associated with more TEs when taking psychedelics in the Frontiers paper).

Apologies on the liberalism comment - I'd mis-read the scale direction.

Happy to see this progressive article take further shape. The extensive comments were only because I found it interesting and, hopefully, were useful rather than feeling too critical!

Best wishes
Martha Newson

Note: reviewer responses (in full) are in bold; our responses are in normal typeface, excerpts in italic. References appear as endnotes at the end of this letter.

Reviewer #1

The authors did a great job in addressing all of my questions and concerns. I continue to have a very positive assessment of this work, and I think it will have a major impact on the field. I think it is ready for publication.

Thank you very much for this positive assessment! We are excited to see the manuscript in print.

Reviewer #2

This was a thorough revision which I felt addressed my previous concerns! I only have one further query. The authors say (ln 406) It is important to note that universal connectedness and the group identity fusion scales differed in their extremity, with the latter indicating —total immersion of the self in the group, while the former indicated only a high degree of overlap, so these measures cannot be compared directly.

This suggests they used the fusion measure as a dichotomous measures (fused=completely overlapping vs. all other partial overlap ratings)? However, this was not clear in the methods, and rather it sounded like they were treating it as a continuous measure? Theoretically it should be treated as dichotomous, and perhaps that is how it was treated, but I could not see where that was specified?

Thank you for your question about the comparison of the universal connectedness versus and group identity fusion measures. We had added this point of clarification in response to Reviewer 3's concern articulated in the previous round of revision:

Fusion is not just relationally different but different in its depth, it is a more visceral and enduring bonding – so the final option of fusion (total immersion) is categorically different from connectedness measured by the IOS.
—R3

We understood this concern to be in reference to the nature of the two measures. As you can see in the images below, the highest endpoint of the “universal connectedness” measure has almost-completely-overlapping circles, while the “group identity fusion” measure has one circle completely subsumed in the other. Therefore, as R3 correctly noted, direct comparison between the means of these measures is limited. While our analysis relied on regression to examine “variance explained” between these measures, and therefore largely avoids problems of direct comparison, we nevertheless agreed that this aspect of the differences between the measures should be pointed out to readers. We certainly did not mean to convey the impression that the universal connection measure was treated as dichotomous, which would have significantly reduced the precision of the measure. Moreover, given that our theoretical approach is designed

to capture degrees of change in connectedness and prosocial orientation, we consider the continuous measure to be the more appropriate estimate of the construct.

We agree that we did not do an adequate job of conveying this aspect of the measure in our Methods section. We have altered the text accordingly (p. 22):

This measure was scored on a continuous scale from 1 to 7.

Universal connectedness:

Please circle the image that best describes your current relationship with other human beings, in general.

Group identity fusion:

Please circle the image below that you feel best represents your relationship with other Burners as a group.

Reviewer #3

You've addressed all of my queries well. I would recommend Newson et al., 2021 (Frontiers - for transparency, this is my paper!), which investigates the role of psychedelics on transformative experiences (which in turn appeared to be associated with identity fusion). This might be useful with the Frostman reference showing that work has already been done on this area, hence your paper not focussing on drugs. It may also be relevant to your new lines on personality traits (trait openness was associated with more TEs when taking psychedelics in the Frontiers paper).

Thank you for the reference! In fact, we were thrilled to see this paper emerge a few months ago and had included it as citation #7, and briefly referenced some of its (very relevant!) results in the penultimate paragraph of the manuscript.

As we wrote:

Finally, it is important to note that while our research focuses on transformative experiences at multi-day secular mass gatherings, this is by no means the only environment where such experiences can occur. Apart from psychedelic experiences,

these include interacting with literature and music , practicing meditation , or immersing oneself in nature . Given the diversity of these settings, it remains unclear exactly which aspects of mass gathering attendance cause transformative experiences. While our study was not primarily designed to answer this question, our analyses suggest that activities such as dancing (which can elicit emotional synchrony^{18,22,30}), giving and receiving gifts (which may amplify prosocial emotions), and making new friends (which may increase feelings of mutual obligation^{9,8,10}) partially mediated the relationship between time onsite and transformative experience (SOM 1.13 for more details). Other research has identified sleep deprivation and exposure to powerful rhythmic music as additional causes of transformative experience⁷. Additional research will be needed to better understand how these factors interact with aspects of the local event culture and individual differences to produce prosocial transformation.

Thank you for your excellent work on this topic!

Apologies on the liberalism comment - I'd mis-read the scale direction.

No problem!

Happy to see this progressive article take further shape. The extensive comments were only because I found it interesting and, hopefully, were useful rather than feeling too critical!

We were sincerely grateful for all your helpful directions and believe the paper is in the best shape it's ever been in, thanks in large part to your feedback!